# Functional visualization of NK cell-mediated killing of metastatic single tumor cells

**Hiroshi Ichise[1], Shoko Tsukamoto[1], Tsuyoshi Hirashima[2], Yoshinobu Konishi[1], Choji Oki[3], Shinya Tsukiji[3], Satoshi Iwano[4], Atsushi Miyawaki[4], Kenta Sumiyama[5], Kenta Terai[1], Michiyuki Matsuda[1,2,6]\***

[1]Research Center for Dynamic Living Systems, Graduate School of Biostudies, Kyoto University, Kyoto, Japan; [2]Department of Pathology and Biology of Diseases, Graduate School of Medicine, Kyoto University, Kyoto, Japan; [3]Department of Nanopharmaceutical Sciences, Nagoya Institute of Technology, Gokiso-cho, Nagoya, Japan; [4]Brain Science Institute, Center for Brain Science, RIKEN, Hirosawa, Wako, Japan; [5]Laboratory for Mouse Genetic Engineering, RIKEN Center for Biosystems Dynamics Research, Suita, Japan; [6]Institute for Integrated Cell-Material Sciences, Kyoto University, Kyoto, Japan

**\*For correspondence:**
matsuda.michiyuki.2c@kyoto-u.ac.jp

**Competing interest:** The authors declare that no competing interests exist.

**Abstract** Natural killer (NK) cells lyse invading tumor cells to limit metastatic growth in the lung, but how some cancers evade this host protective mechanism to establish a growing lesion is unknown. Here, we have combined ultra-sensitive bioluminescence imaging with intravital two-photon microscopy involving genetically encoded biosensors to examine this question. NK cells eliminated disseminated tumor cells from the lung within 24 hr of arrival, but not thereafter. Intravital dynamic imaging revealed that 50% of NK-tumor cell encounters lead to tumor cell death in the first 4 hr after tumor cell arrival, but after 24 hr of arrival, nearly 100% of the interactions result in the survival of the tumor cell. During this 24-hr period, the probability of ERK activation in NK cells upon encountering the tumor cells was decreased from 68% to 8%, which correlated with the loss of the activating ligand CD155/PVR/Necl5 from the tumor cell surface. Thus, by quantitatively visualizing, the NK-tumor cell interaction at the early stage of metastasis, we have revealed the crucial parameters of NK cell immune surveillance in the lung.

## Editor's evaluation

This is an interesting study that significantly contributes to our understanding of the immunological mechanisms that limit cancer metastasis. Specifically, the authors make use of advanced imaging techniques to characterize the earliest responses of NK cells against tumor cells in cancer metastasis.

## Introduction

Natural killer (NK) cells are innate lymphoid cells that play critical roles in protecting against the development of tumor metastases (*Chiossone et al., 2018*). In human patients, a higher number of circulating or tumor-infiltrating NK cells is correlated with better patient outcomes (*López-Soto et al., 2017*). In immunocompetent mice, selective depletion of NK cells markedly increases the metastatic burden (*Diefenbach et al., 2001*; *Smyth et al., 1999*). Most of these previous studies relied on the number of macro-metastatic tumors as a single functional endpoint, preventing insight into the step(s) of the metastatic cascade at which NK cells play the most critical role. Metastasis involves

the migration of a tumor cell or tumor cell cluster from the primary cancer site through the blood, lodging of the migrating cells in a micro-vessel, and the transmigration of the cell(s) into the tissue parenchyma, where they may either remain dormant or grow into a larger tumor mass (*Gupta and Massagué, 2006*). Given this sequence of events, it is unknown where NK cell attack on the malignant cells giving rise to a metastatic lesion takes place. Using a newly developed method of intravascular staining of immune cells (*Anderson et al., 2014*), it was shown that more than 90% of NK cells in the mouse lung are in the vasculature (*Secklehner et al., 2019*). In agreement with this finding, the major NK cell subset in the lung is similar to that in the peripheral blood (*Hayakawa and Smyth, 2006*; *Marquardt et al., 2017*). Meanwhile, it has also been proposed that NK cells in the normal lung are incompetent or hypofunctional (*Marquardt et al., 2017*; *Robinson et al., 1984*). Thus, it remains unclear how pulmonary NK cells are able to prevent tumor cells from colonization in the lung.

Inhibition of platelets and coagulation factors has long been associated with the suppression of lung metastasis (*Brown, 1973*; *Gasic et al., 1968*; *Pearlstein et al., 1984*). At least a part of the pro-metastatic effect of the coagulation cascade is attributed to inhibition of NK cells (*Gorelik et al., 1984*; *Nieswandt et al., 1999*; *Palumbo et al., 2005*; *Sadallah et al., 2016*). But other mechanisms such as enhanced adhesion of tumor cells (*Nierodzik et al., 1992*), formation of a favorable intra-vascular metastatic niche (*Lucotti et al., 2019*), or TGF-β1-mediated immune evasion (*Metelli et al., 2020*) have also been proposed. Therefore, a method to untangle the metastatic cascade in vivo is needed to quantitatively assess the effect of anticoagulants on each step and the possible relationship of this anti-metastatic action to the function of NK cells.

Intravital two-photon (2P) microscopy enables visualization of the interaction of immune cells with other immune cells or their targets (*Cyster, 2010*; *Germain et al., 2012*; *Liew and Kubes, 2015*). For example, the dynamic behaviors of NK cells have been demonstrated in the lymph nodes (*Bajénoff et al., 2006*; *Beuneu et al., 2009*; *Mingozzi et al., 2016*) and the tumor microenvironment (*Barry et al., 2018*; *Deguine et al., 2010*). Moreover, the development of genetically encoded biosensors for signaling molecules has paved the way to monitoring of cellular activation status in not only normal but also pathological tissues (*Conway et al., 2017*; *Terai et al., 2019*).

Here, we have combined bioluminescence whole-body imaging and intravital 2P microscopy to explore the behavior and functional competence of NK cells in an experimental lung metastasis model. Using an ultra-sensitive bioluminescence system, we followed the fate of intravenously injected tumor cells from 5 min to 10 days. The number of viable disseminated tumor cells in the lung decrease rapidly and reach a nadir within 12–24 hr in an NK cell-dependent manner. Intravital 2P microscopy demonstrates that a static tumor cell in a pulmonary capillary is contacted by a crawling NK cell approximately every 2 hr. Importantly, the probability of NK cell activation and subsequent elimination of the lodged tumor cell decreases rapidly after 24 hr of arrival in the lung capillary bed. We show that this evasion of NK cell surveillance is inversely correlated with thrombin-dependent shedding of CD155/PVR/Necl5 (hereafter simply Necl5), a ligand for the NK cell-activating receptor DNAM-1. This loss of surface activating ligand limits the signaling needed to invoke NK cytotoxicity, thus protecting the tumor cell and enabling the formation of a growing metastatic lesion. Anticoagulants promote tumor killing by NK cells by limiting this loss of activating ligand.

## Results

### NK cells eliminate disseminated tumor cells within 12–24 hr after the entry into the lung

Development of an extremely bright bioluminescence imaging system, AkaBLI (*Iwano et al., 2018*), enabled us to visualize the acute phase of lung metastasis and to explore the role of NK cells in the elimination of disseminated tumor cells. B16F10 melanoma cells were transduced with Akaluc luciferase, and the resulting cells, called B16-Akaluc cells hereafter, were injected intravenously into syngeneic C57BL/6 mice that had previously been injected with either an anti-asialo GM1 (αAGM1 hereafter) antibody or an isotype control antibody. The pretreatment with αAGM1 removed more than 97% of NK cells from the spleen and the lung (*Figure 1—figure supplement 1A and B*). Immediately after the intravenous injection of tumor cells, we administered AkaLumine luciferin intraperitoneally (i.p.) and started bioluminescence imaging under anesthesia (*Figure 1A* and *Figure 1—video 1*). A bioluminescence signal in the control mice was observed almost exclusively in the lung and decreased

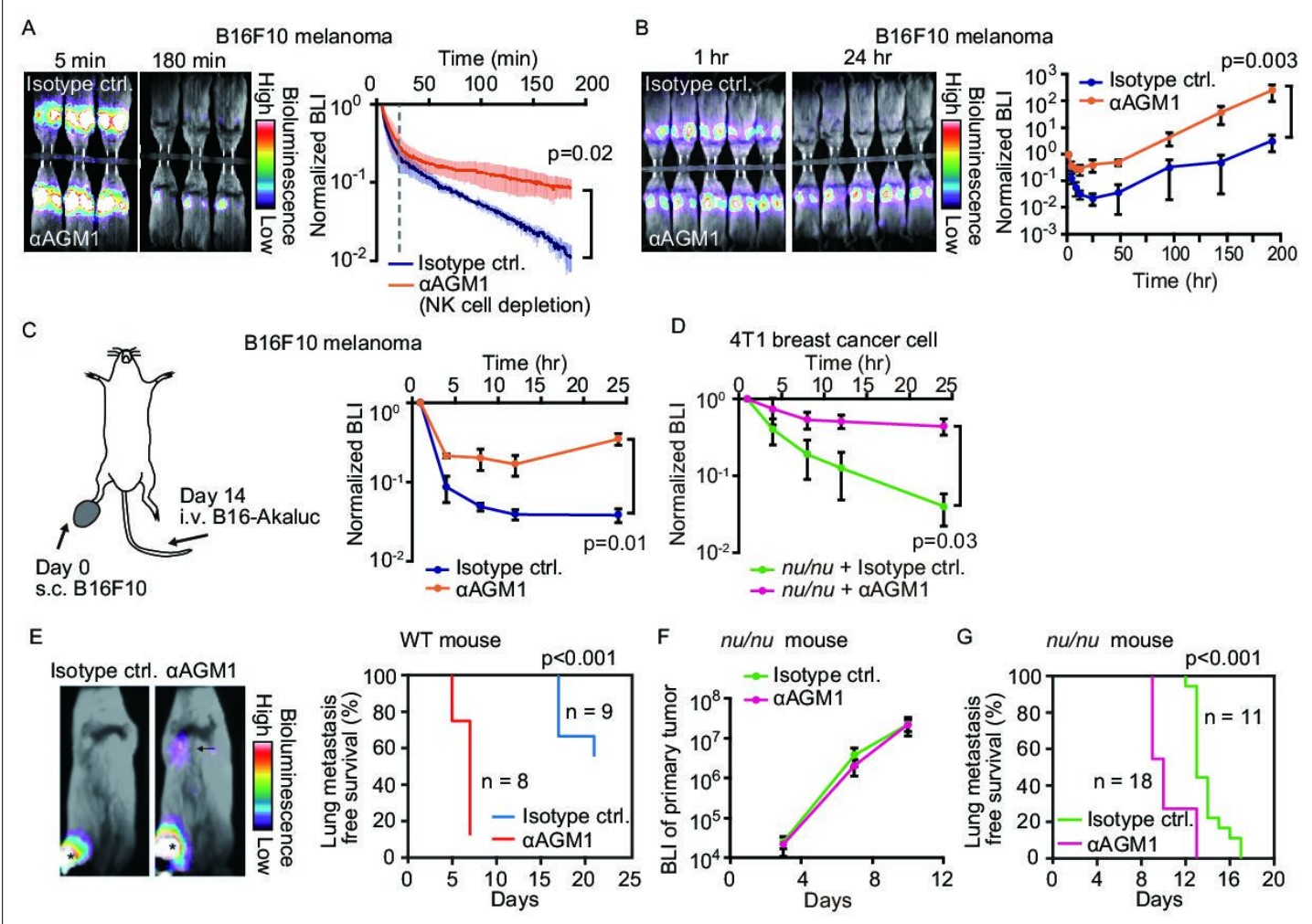

**Figure 1.** NK cells eliminate a subset of metastatic tumor cells from the lung within 24 hr of arrival. (**A, B**) B6 mice were pretreated with either control antibody or αAGM1. Representative merged images of the bright field and the bioluminescence images of mice intravenously injected with 5×10⁵ B16-Akaluc cells are shown. (**A**) The substrate was i.p. administered immediately after injection of tumor cells. Image acquisition was started at 5 min after tumor injection. See also *Figure 1—video 1*. Bioluminescence intensity (BLI) is normalized to that at 5 min and plotted over time. Data are representative of two independent experiments with three mice per group and are shown as means ± SD. A dotted line represents 20 min. (**B**) Substrate was administered i.p. before each round of image acquisition. BLI is normalized to that at 1 hr. Data are representative of three independent experiments with 4–6 mice per group and shown as means ± SD. (**C**) B16-Akaluc cells were injected into the tail vein of mice that had been inoculated with B16F10 cells in the footpad 14 days before. BLI was quantified at the indicated time and normalized to that at 1 hr after injection of the B16-Akaluc cells. Data are representative of two independent experiments with three mice per group and are represented as means ± SD. (**D**) Nude mice were pretreated with either control antibody or αAGM1. 4T1-Akaluc cells were injected into the tail vein. BLI was quantified at the indicated time and normalized to that at 1 hr after injection of the tumor cells. Data are representative of two independent experiments with four mice per group and are represented as means ± SD. (**E**) BALB/c mice were pretreated with either control antibody or αAGM1. Shown are representative merged images of the bright field and the bioluminescence images of mice subcutaneously injected with 5×10⁵ 4T1-Akaluc cells into footpad. An arrow and asterisks depict a lung metastasis and primary tumors, respectively. Lung metastasis incidence of control antibody- (n=9) or αAGM1- (n=8) treated mice. (**F, G**) Identical to (**E**) except that mice are nude mice and the implanted cell number is 1×10⁴. (**F**) BLI of primary tumor. Data are representative of two independent experiments with three mice per group and are represented as means ± SD. (**G**) Lung metastasis incidence of control antibody- (n=11) or αAGM1- (n=18) treated mice. NK, natural killer.

The online version of this article includes the following video and figure supplement(s) for figure 1:

**Figure supplement 1.** Depletion of NK cells by αAGM1.

**Figure supplement 2.** NK cells eliminate metastatic tumor cells from the lung within 24 hr.

**Figure supplement 3.** Basophils, macrophages, and neutrophils do not contribute to elimination of metastatic tumor cells.

**Figure 1—video 1.** Acute rejection of metastatic tumor cells by NK cells.

https://elifesciences.org/articles/76269/figures#fig1video1

rapidly in the first 20 min, then gradually thereafter. In the αAGM1-treated mice, the biolumines-cence signal dropped rapidly as observed in the control mice; however, the decrease was substantially reduced after 20 min as compared to control animals. Thus, in the initial phase (<180 min), there are at least two mechanisms that eliminate melanoma cells from the lung. The rapid elimination of mela-noma cells (<20 min) may reflect flushing away by the blood flow or shear-stress-mediated cell death. The slow component of the elimination (>20 min) observed in the control mice is caused primarily by NK cells.

To explore the NK cell-mediated elimination of tumor cells after the early rapid phase decline (1 hr to 8 days), we administered luciferin immediately before each round of imaging (*Figure 1B*). The bioluminescence signals were normalized to that at 1 hr after B16-Akaluc injection in each mouse. In the control mice, the bioluminescence signal of melanoma cells reached a nadir 24 hr after tumor cell injection and increased thereafter, indicating proliferation of melanoma cells. On the other hand, in the αAGM1-treated mice, the bioluminescence signal decreased very little after 4 hr and started increasing after 12 hr. Importantly, after 24 hr, we did not observe any significant difference in the relative increase of the bioluminescence signal between the control and αAGM1-treated mice. In both mice, the doubling time of melanoma cells is approximately 1 day, implying that NK cells elim-inate disseminated melanoma cells primarily in the acute phase (<24 hr) of lung metastasis. In tumor burden mice, the lung microenvironment is often reprogrammed in favor of metastasis (*Altorki et al., 2019*). To test this possibility, B16F10 melanoma cells were inoculated into the footpad 2 weeks prior to the intravenous injection of the B16-Akaluc cells (*Figure 1C*). In this model, B16F10 melanoma cells continue to grow at the footpad until the day of B16-Akaluc cell injection. αAGM1 treatment hampered the rapid decrease of the bioluminescence signal, suggesting that NK cell-mediated elimi-nation of metastatic melanoma cells operates in the melanoma-burdened mice as well.

We extended this approach to other syngeneic mouse tumor cell lines: Braf$^{V600E}$ melanoma cells (*Dhomen et al., 2009*), MC-38 colon adenocarcinoma cells (*Rosenberg et al., 1986*), and BALB/c mice-derived 4T1 breast cancer cells (hereinafter called 4T1-Akaluk). The rapid decrease of biolu-minescence signals was markedly alleviated by αAGM1, supporting the critical role of NK cells in the acute phase (*Figure 1—figure supplement 2A–C*). It has been reported that the αAGM1 reacts with basophils (*Nishikado et al., 2011*). Therefore, we repeated the experiment using an αCD200R3 basophil-depleting antibody (Ba103). The basophil depletion did not affect the elimination of mela-noma cells (*Figure 1—figure supplement 3A and B*). Similarly, the roles of circulating monocytes and neutrophils were examined with clodronate liposome and αLy-6G neutrophil antibody, respectively. Neither treatment mitigated the melanoma elimination within 24 hr of injection (*Figure 1—figure supplement 3C–F*). Although the effect of these reagents to eliminate monocytes and neutrophils was not complete, these data suggest the involvement of monocytes and neutrophils in the rejection of melanoma cells in the acute phase.

The effect of T cell immunity on tumor elimination was examined by using *Foxn1$^{nu/nu}$* mice (here-inafter called nude mice). 4T1-Akaluc cells were injected into nude mice (*Figure 1D*). 4T1-Akaluc cells were eliminated in mice as efficiently as in wild-type (WT) mice in an NK cell-dependent manner, indicating that under these conditions, T cell immunity does not contribute to tumor cell reduction in the acute phase of rejection (<24 hr). To further explore the NK cell-mediated tumor cell elimination in a spontaneous metastasis model, 4T1-Akaluc cells were inoculated into the footpad of WT and nude mice (*Figure 1E–G*). The αAGM1 treatment did not affect the growth of the primary tumor, but suppressed lung metastasis (*Figure 1E and G*). These results indicate the critical role of NK cells in the elimination of disseminated tumor cells. The experiment with nude mice also excluded the involve-ment of NKT and γδ T cells, with which αAGM1 reacts (*Trambley et al., 1999*). Of note, ILC1 cells also react with αAGM1. But, ILC1 is known to promote lung metastasis of B16F10 cells (*Gao et al., 2017*), suggesting that the effect of αAGM1 is primarily caused by the depletion of NK cells.

## NK cell dynamic behavior in pulmonary capillaries

To better understand how NK cells mediate rapid elimination of metastatic tumor cells, we examined their topographic distribution and dynamics in the lung. We first assessed the localization of NK cells by intravascular staining with αCD45 antibody (*Anderson et al., 2014*), followed by flow cytom-etry with αCD45, αCD3, and αNCR1 antibodies (*Figure 2—figure supplement 1*). Consistent with previous reports (*Gasteiger et al., 2015*; *Secklehner et al., 2019*), more than 95% of pulmonary NK

cells (CD45[+], CD3[−], and NCR1[+]) were found in the vasculature, compared to 20% of bone marrow NK cells. To study the dynamics of NK cell-mediated immune surveillance in the lung, we developed a reporter line, NK-tdTomato mice, whose NK cells, or more specifically, NCR1[+] cells (*Narni-Mancinelli et al., 2011*), express the fluorescent protein tdTomato. Next, we employed in vivo pulmonary imaging by 2P microscopy to observe NK cells in situ for up to 12 hr (*Figure 2A*). In agreement with the flow cytometric data, most NK cells were found within the capillaries (*Figure 2B*). NK cells flowing in the capillaries stalled on endothelial cells, crawled a short distance, and jumped back into the flow (*Figure 2C* and *Figure 2—video 1*). A histogram of the crawling time exhibited exponential decay with a median of 2.5 min (*Figure 2D*). NK cells are known to express at least two integrin-family proteins, LFA-1 (LFA-1α/CD18 and αLβ2) and Mac-1 (Mac-1/CD18 and αMβ2) (*Wang et al., 2012*). Intravenous injection of the blocking antibody against LFA-1α (αLFA-1α) resulted in a reduction of NK cells on the pulmonary endothelial cells (*Figure 2E*). To extend this observation, we counted the number of NK cells on the endothelial cells in the presence or absence of αLFA-1α. As expected, αLFA-1α, but not αMac-1, markedly reduced the number of NK cells on the pulmonary endothelial cells (*Figure 2F*), indicating that the adhesion of NK cells to the pulmonary endothelial cells is mediated at least partially by LFA-1.

## Intra-pulmonary NK cell patrolling and tumor interaction dynamics

To understand better the kinetics of NK cell migration within pulmonary vessels and their interactions with lodged tumor cells, we examined the three-dimensional trajectory of the crawling NK cells in the presence and absence of B16F10 cells (*Figure 3A–C*). In this experiment, we used B16-SCAT3 cells, which expressed the Förster resonance energy transfer (FRET)-based caspase-3 biosensor SCAT3 (*Takemoto et al., 2003*). The ratio of NK cells versus tumor cells in the field of view (FOV) was set approximately 1:1. We compared the mean square displacement (MSD) of NK cells in the presence versus absence of tumor cells (*Figure 3D*). Curves fitted with the measured data represent α<1 regardless of the presence of tumor cells, α=0.56 in the presence and α=0.53 in the absence, indicating that the migration mode of NK cells can be classified as sub-diffusive. This is probably because the migration of NK cells is limited to the pulmonary vascular structure. MSD analysis also shows that the displacement from the original position is smaller in the presence than in the absence of tumor cells. This observation is consistent with our present finding that the mean instantaneous speed of the crawling NK cells was significantly decelerated in the presence of tumor cells, from 7.8 to 4.8 μm/min (*Figure 3E*). In accordance with this observation, the median duration time of crawling was markedly increased in the presence of tumor cells, from 5 to 30 min (*Figure 3F*). Notably, these data exclude regions in which the tumor cells would physically block NK movement in the capillary, suggesting that the dissemination of tumor cells causes global activation of pulmonary endothelial cells, and thereby causes slower crawling of NK cells. Taking advantage of these quantitative imaging data, we summarized parameters regarding the dynamics of pulmonary NK cells (*Table 1*). The parameters on flowing NK cells were deduced from the NK cell count in blood and the flow rate in pulmonary capillaries. In short, a tumor cell lodged in a pulmonary capillary will be contacted by the flowing and crawling NK cells roughly every 10 min and every 2 hr, respectively.

## Necl5 and Nectin2 on tumor cells stimulate NK cell signaling leading to tumor cell killing

During serial killing of tumor cells by NK cells, perforin/GrzB initially plays a major role, followed by Fas-mediated killing (*Prager et al., 2019*). Therefore, we used two biosensors to detect NK cell-mediated killing in vivo: SCAT-3 for caspase activation by any pathway and GCaMP-6s for specific detection of the perforin/GrzB-mediated membrane damage. For the typical B16-SCAT3 cells, caspase-3 was activated 16 min after an NK cell came into contact with the tumor cell (*Figure 4A and B* and *Figure 4—video 1*). The caspase-3 activation was observed in 18% of contact events within a median of 26 min after the contact (*Figure 4C and D*). To examine the possible cause of this limited extent of measurable cell death after contact with an NK cell, we studied Ca[2+] influx in tumor cells, which is known to herald apoptosis (*Keefe et al., 2005*), by using two Ca[2+] sensors, GCaMP6s (*Chen et al., 2013*) and R-GECO1 (*Zhao et al., 2011*). In preliminary in vitro experiments, the B16F10 cells expressing R-GECO1 (B16-R-GECO) were co-cultured with NK cells that had been activated by IL-2 in vitro. Typically, Ca[2+] influx was observed within a few min after contact (*Figure 4—figure*

**eLife** Research article

Cell Biology | Immunology and Inflammation

**Figure 2.** NK cells patrol pulmonary capillaries in a stall-crawl-jump manner. (**A**) A schematic of the intravital imaging system for the lung. The left lobe of the lung was exposed by 5th or 6th intercostal thoracotomy using custom-made retractors and fixed to the objective by a vacuum-stabilized imaging window. (**B**) (Left) A micrograph of the lung of an NK-tdTomato mouse, in which NK cells express tdTomato (magenta). Lectin (green) was injected intravenously to stain endothelial cells. (Right) A magnified image of the boxed region in the left panel. (**C**) A representative time-lapse image of NK cells (magenta). The track of an NK cell is shown with a cyan dotted line and white arrowheads at both ends. White and yellow dotted circles show the positions of stall and jump, respectively. See also *Figure 2—video 1*. (**D**) Distribution of the crawling duration times in a 0.25-mm² field of view (FOV). Data are pooled from two independent experiments (n=583). (**E, F**) B6 mice expressing tdTomato in NK cells (magenta) were observed by 2P microscopy. During time-lapse imaging with a 30-s interval, 100 µg of αLFA-1α, αMac-1, or isotype control antibody was intravenously injected. The number of NK cells in the 0.25-mm² FOV was counted 0–10 min before and 30 min after antibody injection. The percentage of NK cells after versus before antibody injection is shown in (**F**). Data were pooled from three independent experiments and represented as means ± SD. n=3 mice for each group. 2P, two-photon; NK, natural killer.

The online version of this article includes the following video and figure supplement(s) for figure 2:

**Figure supplement 1.** Intravascular staining of NK cells.

**Figure 2—video 1.** Stall-crawl-jump movement of NK cells in the pulmonary capillary.

https://elifesciences.org/articles/76269/figures#fig2video1

supplement 1A and B). A surge of Ca²⁺ influx was observed only in cells that were doomed to die (*Figure 4—figure supplement 1C* and *Figure 4—video 2*); 98% of cells that exhibited Ca²⁺ influx died by apoptosis with blebbing (Figure S5D). With these in vitro data in hand, we used the surge of Ca²⁺ influx as the surrogate marker for apoptosis of metastatic melanoma cells in vivo. In a typical example

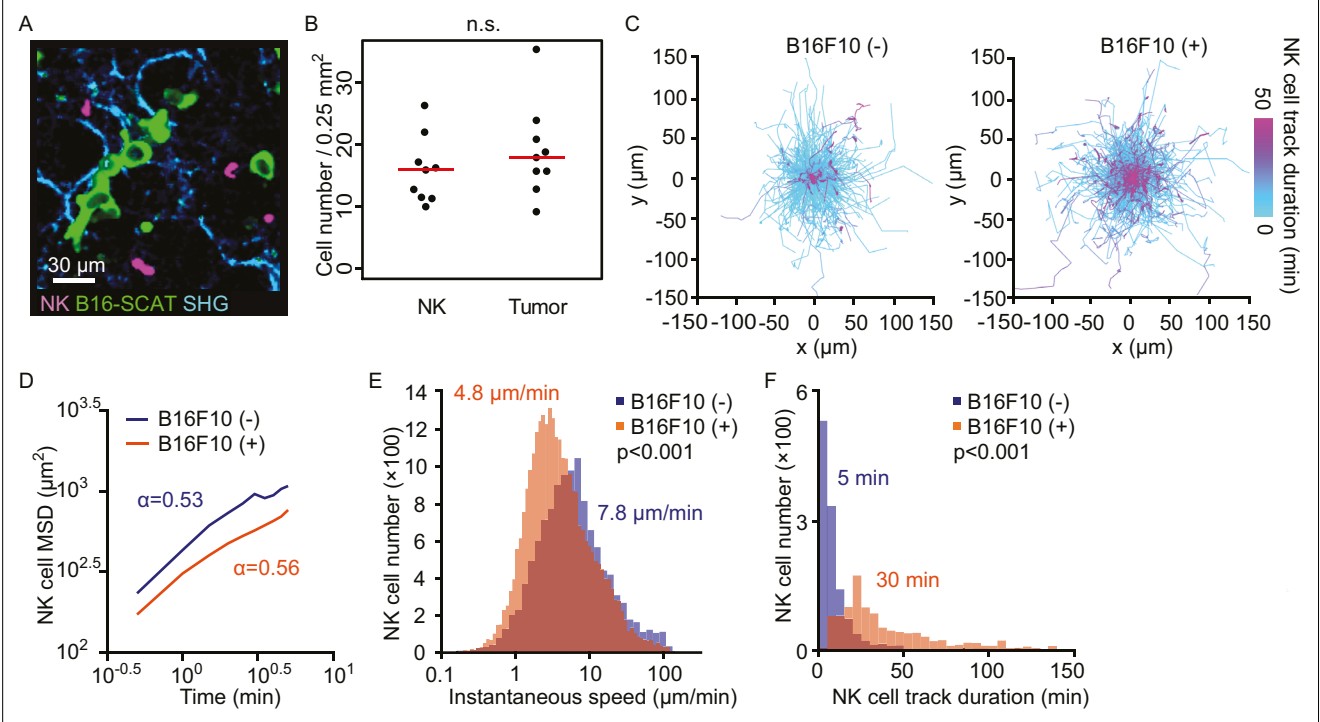

**Figure 3.** NK cells patrol capillaries deliberately in the presence of melanoma. NK-tdTomato mice were injected with 1.5×10⁶ B16-SCAT3 cells and observed under a 2P microscope for 2 hr from 6 hr after injection. NK-tdTomato mice without any treatment were used as the control. SHG stands for second-harmonic generation. (**A**) A micrograph of the lung of an NK-tdTomato mouse. B16-SCAT3 cells, green; NK cells, magenta. (**B**) The average number of NK cells and tumor cells in each FOV at 10–35 μm from the pleura 10 min after intravenous injection of B16F10 cells (n=9). The red lines represent the mean. (**C**) Trajectories of crawling NK cells in the presence (left) or absence (right) of B16F10 cells. For 3D tracking, images of a 0.25-mm² FOV and 25 μm thickness at 10–35 μm from the pleura were acquired every 30 s for 120 min. Shown here are the trajectories of NK cells projected onto the XY plane. Each track is shown in pseudo-color based on track duration. Data are at least from two independent experiments for each condition. n=1127 cells in the absence and n=718 cells in the presence of tumor cells. (**D–F**) Shown are mean squared displacement (MSD) (**D**), instantaneous speed (**E**), and track duration (**F**). The statistical differences between the two experimental groups were assessed by Mann–Whitney U-test. 2P, two-photon; FOV, field of view; NK, natural killer.

(*Figure 4E and F* and *Figure 4—video 3*), when an NK cell contacted a B16F10 cell expressing GCaMP6s (B16-GCaMP), Ca²⁺ influx was induced within 3 min. The Ca²⁺ influx was observed with a median lag of 6 min (*Figure 4G*) in 47% of contact events (*Figure 4H*). These data suggest that the reason why NK cells failed to induce caspase-3 activation in four-fifths of the tumor cells may be due to the limitation of observation period after the delivery of the lethal hit. In addition, the half-life of B16F10 cells based on the probability of an NK cell killing and crawling speed is 137 min whereas the one calculated based on the bioluminescence intensity (BLI) is 146 min (*Table 1*), suggesting that Ca²⁺ influx well correlates with BLI. It is reported that DNAM-1 on NK cells contributes to the elimination of B16F10 melanoma cells (*Gilfillan et al., 2008*). Therefore, to connect these data on tumor cell death with NK cell activation, we used B16F10 cells deficient in expression of CD155/PVR/Necl5 and CD112/Nectin2 and (Necl5 and Nectin2, hereafter), the ligands for the activating receptor DNAM-1 on NK cells (*Chan et al., 2010*). As anticipated, tumor Ca²⁺ influx was almost completely abolished in the *Necl5⁻/⁻ Nectin2⁻/⁻* B16F10 cells (*Figure 4H*), suggesting that damage to tumor cells is dependent on the engagement of Necl5 and/or Nectin2 on melanoma cells.

With these data in hand, we could return to the question of whether flowing or crawling NK cells are responsible for tumor cell death. Crawling NK cells accounted for 77% of Ca²⁺ influx events in the melanoma cells (*Figure 4I*). To reveal the impact of the crawling NK cells, αLFA-1α was used to inhibit NK cell attachment to the pulmonary capillaries. This markedly attenuated the melanoma elimination not only within 24 hr, but also after 10 days (*Figure 4J and K*). Our finding suggests that LFA-1 is required for the crawling of NK cells on the endothelial cells, and, thereby, for the immune surveillance against metastatic tumor cells. B16F10 cells do not express the LFA-1 ligands, ICAM-1 and ICAM-2

**Table 1.** Dynamics of NK cell killing of melanoma cells in the lung.

| Symbols | Parameters | Values | Units | References, equations, and comments |
|---|---|---|---|---|
| **Histological and physiological parameters from published papers** | | | | |
| Bv | Blood volume | 1.7E−6 | m$^3$ | Table II (*Davies and Morris, 1993*) |
| Lv | Lung volume | 3.7E−07 | m$^3$ | *Figure 2A* for Week 8 mice (*Gomes et al., 2019*) |
| CO | Cardiac output | 2.0E−05 | m$^3$/min | Abstract (*Janssen et al., 2002*) |
| Cp_l | Total lung capillary length | 1.1E+03 | m | Result section (*Knust et al., 2009*) |
| **Parameters determined experimentally** | | | | |
| FOV | Field of view | 2.5E+07 | m$^2$ | 0.5×0.5 mm$^2$ |
| Plt_speed | Platelet speed | 0.057 0.95 | m/min mm/s | Determined as described previously (*Sano et al., 2016*). |
| Cp_r | Capillary radius | 3.4E−06 | m | Measured on the images of *Figure 2* |
| NK_d | NK cell diameter | 1.0E−5 | m | Measured on the images of *Figure 2* |
| NK_bl | NK cell count in blood | 5.1E+10 | cells/m$^3$ | Determined for C57BL/6 mice of 8–12 weeks old. |
| NK_FOV | NK cell number in a field of view | 16 | cells/FOV | Measured on the images of *Figure 4* |
| NK_speed | NK crawling speed on capillaries | 4.8E−06 | m/min | Determined with time-lapse images of *Figure 4* |
| NK_hit_obs | Observed NK cell hit probability | 8.0E−3 | cells/min | Determined with time-lapse images of *Figure 4* |
| NK_kill | NK cell killing probability | 0.5 | | Determined with time-lapse images of *Figure 4*. |
| Ml_hl_BLI | Melanoma half-life based on BLI data | 146 | min | From bioluminescence images of *Figure 1* at 4 hr. |
| **Calculated parameters** | | | | |
| Cp_fr | Capillary flow rate | 1.60E−12 | m$^3$/min | Plt_speed*π*Cp_r$^2$ |
| NK_wbl | Whole blood NK cells | 8.7E+04 | cells | NK_bl*Bv |
| NK_density | NK cell density in lung | 6.4E+12 | cells/m$^3$ | NK_FOV/FOV/NK_d |
| NK_lung | Total NK cell in lung | 2.4E+06 | cells | NK_density*Lv |
| NK_in | NK cell influx to lung | 1.0E+06 | cells/min | CO*NK_bl |
| NK_out | NK cell decay constant | 0.42 | /min | NK_in/NK_lung |
| NK_hl | NK cell half-life in the lung | 1.6 | min | ln2/NK_out |
| NK_fr | NK cell flow rate per capillary | 0.11 6.6 | cells/min cells/hr | NK_bl*Cp_fr |
| NK_dcp | NK cell density on capillary | 2.1E+03 | cells/m | NK_lung/Cp_l |
| NK_hit_cr | Crawling NK cell hit probability | 0.010 0.6 | cells/min cells/hr | NK_dcp*NK_speed |
| Ml_t_2P_obs | Melanoma decay constant calculated by 2P imaging | 4.0E−03 | /min | NK_kill*NK_hit_obs |

*Table 1 continued on next page*

*Table 1 continued*

| Symbols | Parameters | Values | Units | References, equations, and comments |
|---|---|---|---|---|
| Ml_hl_2P_obs | Melanoma half-life based on 2P imaging data | 173 | min | ln2/Ml_t_2P_obs |
| Ml_t_2P_crsp | Melanoma decay constant based on crawling speed | 5.1E−03 | /min | NK_kill*NK_hit_cr |
| Ml_hl_2P_crsp | Melanoma half-life based on crawling speed | 137 | min | ln2/Ml_t_2P_crsp |

Basic parameters: Macroscopic and histological data were based on previous papers (**Gomes et al., 2019**; **Janssen et al., 2002**; **Knust et al., 2009**). The speed of platelets in the lung capillaries (Plt_speed) was determined as described previously (**Sano et al., 2016**). The speed of platelets in the pulmonary capillaries was roughly one-third of the speed of platelets in the arteriole of mouse bladder, 3.1 mm/s (0.186 m/min) (**Sano et al., 2016**). The capillary radius, diameter of NK cells, and number of NK cells in a field of view were determined on at least three images. Plt_speed and Cp_r were used to calculate the capillary flow rate (Cp_fr). To determine the total number of NK cells in the blood, 50 μL of blood was collected from the right ventricle of 8- to 12-week-old C57BL/6 mice, lysed in ACK buffer (155 mM/L NH$_4$Cl, 10 mM/L KHCO$_3$, and 0.1 mM/L EDTA), and analyzed by flow cytometry. The CD3$^-$ NK1.1$^+$ cells were counted as NK cells. This number of NK cells is two- to threefold larger than that reported previously by using C57BL/6J mice (**Banh et al., 2012**). The mean crawling speed on the endothelial cells is described in the text related to Figure 3E. The probability of an NK cell hitting a tumor cell was determined by a MATLAB script (Main_191017.m). The probability of an NK cell killing a tumor cell was 0.5, based on the probability of induction of calcium influx in the target B16 melanoma cells (Figure 4H).

Total number of lung NK cells: The total number of NK cells residing in the lung (NK_lung) was estimated from the mean NK cell density (NK_density) and total lung volume (Lv). The diameter of an NK cell (NK_d) was used as the thickness of the image plane. The number of total NK cells, 2.4 million, is markedly larger than the previous values, which ranged from 0.2 to 1 million (**Bi et al., 2017**; **Grégoire et al., 2007**; **Yan et al., 2014**). In previous studies, the whole lungs were lysed to count the blood cell number. It is possible that the recovery rate might have been low due to insufficient tissue lysis. As described in the main text, we observed comparable numbers of tumor cells and NK cells in each FOV, when 1.5×10$^6$ B16-SCAT cells were injected into NK-tdTomato mice, supporting the fidelity of the number of total NK cells determined in this study.

Dynamics of NK cells: Most of the pulmonary NK cells are within the vasculature (Figure 1—figure supplement 3), and the number of pulmonary NK cells overwhelms that of NK cells in the blood. Thus, the total number of lung NK cells can be used as the total number of NK cells in the lung vasculature. NK cell influx into the lung (NK_in) is obtained from cardiac output (CO) and NK cell count in the blood (NK_bl). If all NK cells stay in the lung with equal probability, the apparent transit time in the lung, or NK cell half-life in the lung, is calculated as 1.6 min from NK_in and NK_lung. This value is markedly smaller than the tracking duration period observed in Figures 3 and 4, indicating that the major population of NK cells in the blood go through the lung without adhesion to the endothelial cells. By using the capillary flow rate (Cp_fr) and NK cell count (NK_bl), the NK cell flow rate per capillary (NK_fr) is determined as 0.11 cells/min. Meanwhile, from the total length of capillaries (Cp_l) and the number of NK cells (NK_lung), NK cell density on the capillary (NK_dcp) is determined as 2.1 cells/mm. From NK_dcp and the crawling speed of NK cells, the probability of a tumor cell being hit by crawling NK cells (NK_hit_cr) becomes 0.010 cells/min. This value is approximately one-tenth of the flow rate of NK cells (NK_fr).

Dynamics of disseminated melanoma cells: The BLI signals from 1 to 12 hr (Figure 1B) were fitted with MATLAB using the following equation:

$$BLI = t^{-1.45} \left[ t, hour \right].$$

With this fitting, the decay rate decreases with time. Because we characterized the NK cell interaction with tumor cells between 4 and 8 hr after tumor cell injection, we determined the half-life of B16F-Akaluc cells from 4 hr (Ml_hl_BLI) and obtained 146 min. Meanwhile, from the probability of an NK cell hitting a tumor cell (NK_hit_obs) and the probability of an NK cell killing a tumor cell (NK_kill), the half-life of tumor cells (Ml_hl_2P_obs) becomes 173 min. If we adopt the probability of a tumor cell hit based on the crawling speed of NK cells, the expected half-life of tumor cells (Ml_hl_2P_crsp) becomes 137 min. Considering the precision of parameters obtained from in vivo imaging data, we believe that the half-life of melanoma cells estimated from the 2P microscopy reasonably matched the half-life of melanoma cells determined by BLI.

(*Figure 4—figure supplement 2*), arguing against the possibility that αLFA-1α directly impairs the association of NK cells with B16F10 cells, a conclusion also consistent with a previous study that demonstrated that LFA-1 deficiency in NK cells did not abrogate the in vitro killing capacity against B16F10 cells (*Zhang et al., 2015*).

## Tumor-mediated stimulation of NK cells declines after several hours of pulmonary residence

To track tumor cell induction of NK cell activation, a key step in the killing process, we took advantage of evidence that engagement of LFA-1 or DNAM-1 with their ligands results in the activation of extracellular signal-regulated kinase (ERK) (*Perez et al., 2003*; *Zhang et al., 2015*). We isolated NK cells from transgenic mice expressing a FRET-based ERK biosensor (*Komatsu et al., 2018*), and co-cultured them with B16F10 cells in vitro. Quantification of ERK activity in each NK cell during contact with the tumor cells demonstrated a positive correlation between ERK activation in NK cells and apoptosis in tumor cells (*Figure 5—figure supplement 1A*). Apoptosis was induced in 47% of tumor cells that were in contact with the ERK-activated NK cells (*Figure 5—figure supplement 1B*). On the other hand, apoptosis was never observed in the tumor cells in contact with the NK cells that failed to exhibit ERK activation. In agreement with this observation, an inhibitor for MAPK/ERK

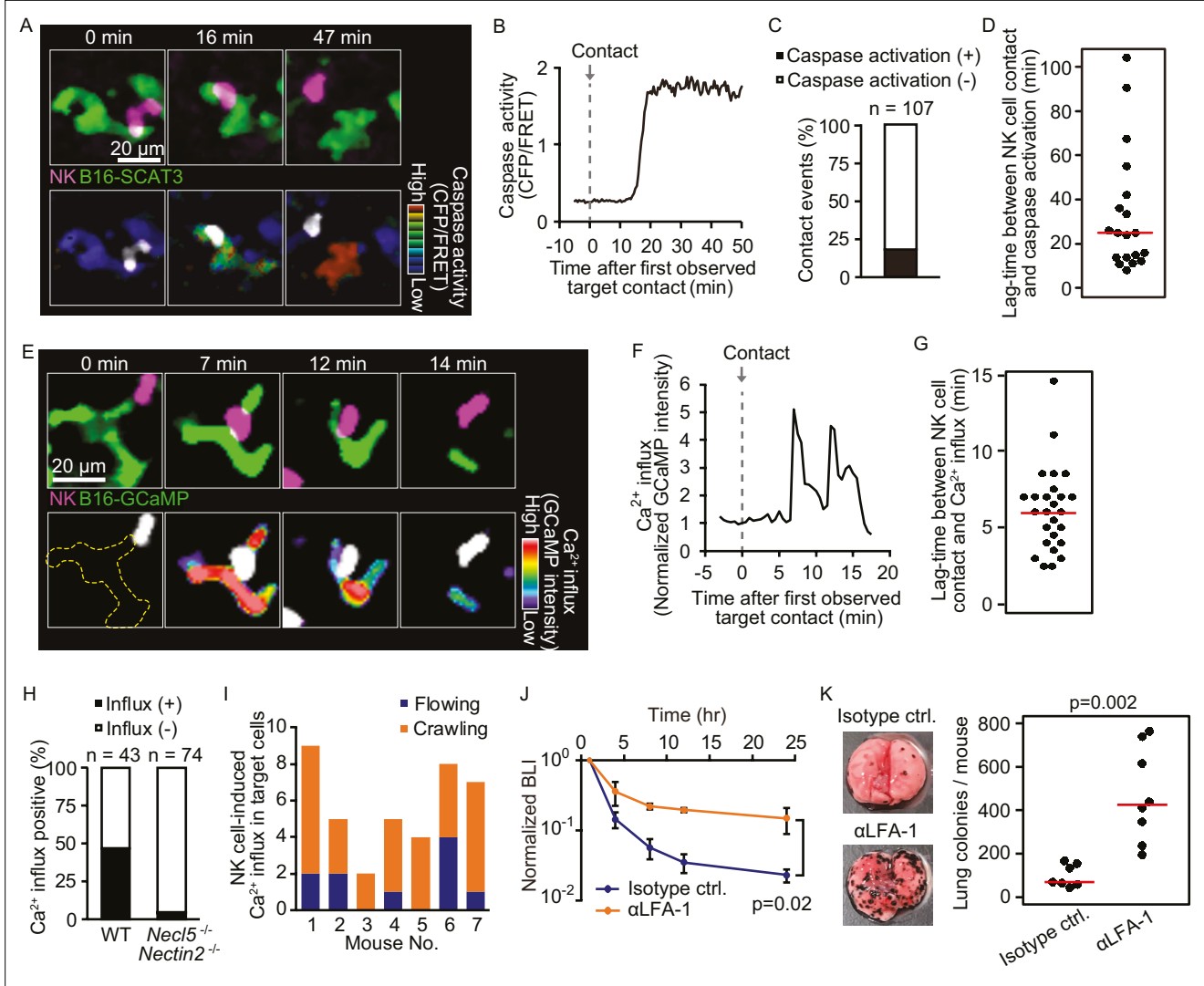

**Figure 4.** Intravital 2P imaging with biosensors visualizes apoptosis and calcium influx of tumor cells induced by crawling, but not flowing, NK cells. (**A**) A representative time-lapse image of a lung of an NK-tdTomato mouse after B16-SCAT3 cell injection. An NK cell and B16-SCAT3 cells are depicted in magenta and green, respectively (top). Bottom, the CFP/FRET ratio in B16-SCAT3 cells is shown in the intensity-modulated display (IMD) mode and an NK cell is shown in white. See also *Figure 4—video 1*. (**B**) Quantification of the CFP/FRET ratio in a B16-SCAT3 cell in (**A**). (**C**) The percentage of NK cell-tumor cell contacts with or without caspase activation. Data were pooled from three independent experiments. (**D**) Time intervals between NK cell contact and caspase three activation in B16-SCAT3 cells. Data are pooled from three independent experiments. (**E**) A representative time-lapse image of a lung of NK-tdTomato mice after B16-GCaMP cell injection. An NK cell and B16-GCaMP cell are depicted in magenta and green, respectively (top). Bottom, GCaMP6s intensity in a B16-GCaMP cell is displayed in pseudo-color and an NK cell is shown in white. See also *Figure 4—video 3*. (**F**) Quantification of GCaMP6s intensity shown in (**E**). (**G**) Time intervals between NK cell contact and Ca$^{2+}$ influx in B16-GCaMP cells. Data were pooled from four independent experiments. Red lines represent the median. (**H**) Comparison of the number of NK cell contacts that were followed by Ca$^{2+}$ influx between the WT and *Necl5$^{-/-}$ Nectin2$^{-/-}$*. Data were pooled from four (WT) and two (*Necl5$^{-/-}$ Nectin2$^{-/-}$*) independent experiments. (**I**) In seven independent experiments, 40 contact events with calcium influx were observed and classified into those caused by crawling or flowing NK cells. (**J**) αLFA-1α or isotype control antibody was intravenously administered 2 hr before injection of 5×10$^5$ B16-Akaluc cells. The bioluminescence signals are normalized to those of 1 hr. Data are representative of two independent experiments with 3–4 mice per group and presented as means ± SD. (**K**) Representative macroscopic images of the metastasis to the lung and number of metastatic nodules per lung are shown. Red lines represent the median. Data were pooled from two independent experiments. Control, n=7; αLFA-1α, n=8. 2P, two-photon; NK, natural killer.

The online version of this article includes the following video and figure supplement(s) for figure 4:

**Figure supplement 1.** NK cell-induced Ca$^{2+}$ influx in B16F10 cells in vitro.

**Figure supplement 2.** Absence of LFA-1 ligands on B16F10 cells.

**Figure 4—video 1.** Induction of caspase three activation by crawling NK cells.

https://elifesciences.org/articles/76269/figures#fig4video1

*Figure 4 continued on next page*

*Figure 4 continued*

**Figure 4—video 2.** Example of morphological changes of a target cell following Ca$^{2+}$ influx mediated by NK cell.
https://elifesciences.org/articles/76269/figures#fig4video2

**Figure 4—video 3.** Induction of calcium influx by crawling NK cells.
https://elifesciences.org/articles/76269/figures#fig4video3

kinase PD0325901, called MEKi hereafter, suppressed NK cell-mediated apoptosis (*Figure 5—figure supplement 1C*). When NK cells were sorted into DNAM-1$^+$ and DNAM-1$^-$, ERK was activated more potently in DNAM-1$^+$ NK cells than in DNAM-1$^-$ NK cells (*Figure 5—figure supplement 1D*). In line with this observation, ERK was not activated in NK cells when B16F10 cells deficient from DNAM-1 ligands were used (*Figure 5—figure supplement 1E and F*). Previously, *Du et al., 2018* reported that IL2-stimulated NK cells poorly recognize *Necl5* and *Nectin2*-deficient B16F10 cells in vitro (*Du et al., 2018*). We confirmed this observation with three independent clones of *Necl5$^{-/-}$ Nectin2$^{-/-}$* B16F10 cell (*Figure 5—figure supplement 1G*). These data support the critical role of DNAM-1 in ERK activation and induction of cytotoxicity at least in vitro.

The link between ERK activation and elimination of metastatic tumor cells was examined in vivo using bioluminescence imaging. At 1 hr before and 8 hr after the injection of B16-Akaluc cells, MEKi or solvent was administered i.p. into mice. MEKi significantly attenuated the rapid decrease of the injected tumor cell number within 24 hr (*Figure 5A*) and the number of lung colonies of MEKi-treated mice was significantly greater than in control mice (*Figure 5B*). At the same time, we observed that MEKi had no additive effect in αAGM1-treated mice in the early (<24 hr) time courses (*Figure 5A*). Although interpretation is limited by the action of the soluble inhibitor on cells other than NK cells, these additional data are consistent with our imaging data and the idea that activated NK cell contributes to the elimination of disseminated tumor cells and that ERK activation can be used as a marker for activated NK cells.

With these data in hand, we next proceeded to visualize ERK activation in vivo. For this, we developed reporter mice whose NK cells express the FRET biosensor for ERK, hereinafter called NK-ERK mice. Intravenous injection of B16-GCaMP cells into NK-ERK mice allowed for simultaneous observation of ERK activity in NK cells and Ca$^{2+}$ influx in melanoma cells. In a representative example, ERK was activated 2.5 min after an NK cell's contact with a melanoma cell (*Figure 5C*, upper panel; *Figure 5D*, magenta), followed by Ca$^{2+}$ influx in the melanoma cells at 5.5 min and cell death at 10 min (*Figure 5C*, lower panel; *Figure 5D*, green). ERK activation, defined by a more than 30% increase in the FRET/CFP ratio, was observed within 3 min in 60 NK cells during 88 contact events (*Figure 5—figure supplement 2A*; *Figure 5E*). Ca$^{2+}$ influx was observed at a median of 4 min in 43 of the 60 tumor cells that came into contact with the NK cell having ERK activation (*Figure 5—figure supplement 2B*; *Figure 5F*). We did not observe Ca$^{2+}$ influx in 28 tumor cells that were touched by the NK cells that failed to show evidence of ERK activation. This result supports the notion that ERK activation in NK cells contributes to the induction of apoptosis in the target tumor cells. Importantly, 24 hr after injection, the probability of ERK activation and Ca$^{2+}$ influx was markedly decreased, indicating NK cells lose the capacity to activate in response to capillary-lodged tumor cells (*Figure 5E and F*).

These results imply that NK cells in the lung are exhausted in 24 hr. Our quantitative data shows that 2.4 million NK cells in the lung outnumber 0.5 million melanoma cells injected into circulation (*Table 1*). Therefore, if NK cell exhaustion could happen, it is not caused by the killing of tumor cells, but by an indirect mechanism such as cytokine-induced suppression or endothelial cell-mediated inactivation. To understand the basis for this loss of NK cell activity by 24 hr after tumor arrival in the lung, B16F10 melanoma cells without Akaluc were injected 24 hr before the injection of B16-Akaluc cells (*Figure 5G*). We did not observe any effect of the pre-injected B16F10 cells on the time course of clearance of the B16-Akaluc cells, indicating that NK cells do not lose tumoricidal activity, but that over time, the tumor cells acquire the capacity to evade NK cell surveillance.

## Thrombin-mediated shedding of Necl5 correlates evasion of NK cell surveillance

DNAM-1-mediated signaling contributes to tumoricidal activity of NK cells, leading us to reason that cell surface expression of the DNAM-1 ligands Necl5 and Nectin2 may diminish over this time period.

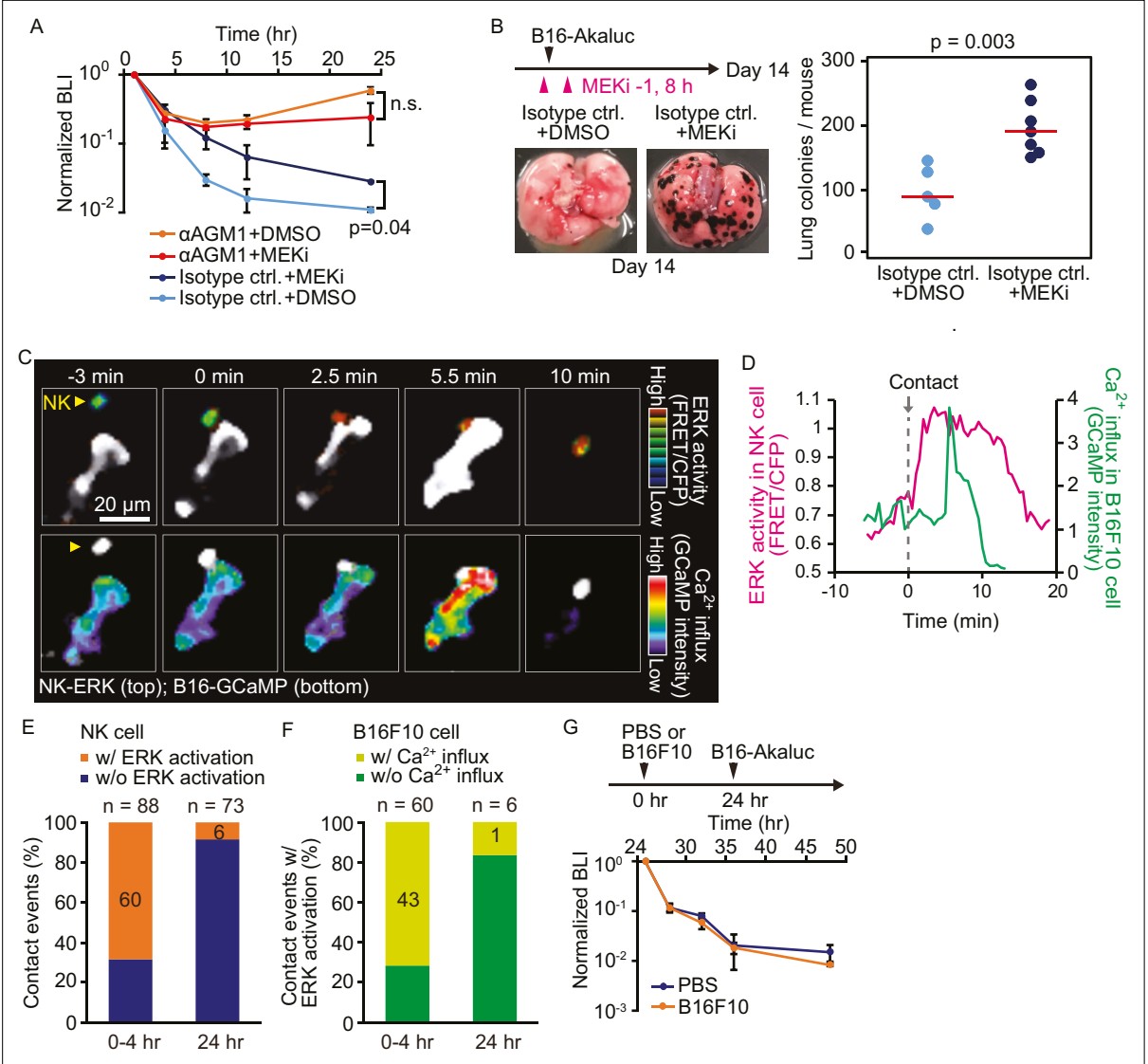

**Figure 5.** Contact-induced ERK activation in NK cells is a necessary event in induction of apoptosis in tumor cells in the first 4 hr, but not after 24 hr. (**A**) B6 mice were pretreated with either control antibody or αAGM1 and intravenously injected with $5 \times 10^5$ B16-Akaluc cells. At 1 hr before and 8 hr after injection, a MEK inhibitor (MEKi) or DMSO was administered i.p. Time courses of the signals, which are normalized to those at 1 hr after tumor injection for each mouse. Data are representative of two independent experiments and shown as means ± SD. n.s., not significant. (**B**) Macroscopic images were acquired at day 14. The number of metastatic colonies is shown. Control, n=5; MEKi, n=7. (**C**) A time-lapse image of the lung of an NK-ERK mouse expressing the FRET biosensor for ERK. The mouse was intravenously injected with B16-GCaMP cells. Top, FRET/CFP ratio of an NK cell (yellow arrowhead) is shown in IMD mode. A B16-GCaMP cell is shown in white. Bottom, GCaMP6s intensity is displayed in pseudo-color. The NK cell is shown in white. (**D**) Time course of the FRET/CFP ratio in the NK cell and CaMP6s intensity in the B16-GCaMP cell. (**E**) Activation probability of ERK in the NK cells upon target cell contact at 0–4 hr or 24 hr after tumor injection. Data were pooled from three independent experiments. (**F**) The probability of NK cells that exhibited ERK activation with or without induction of $Ca^{2+}$ influx in the target tumor cells at 0–4 hr or 24 hr after tumor injection. Data were pooled from three independent experiments. (**G**) B16-Akaluc cells were injected into the tail vein of mice that had been injected PBS or B16F10 cells into tail vein 24 hr before. BLI was quantified at the indicated time and normalized to that at 1 hr after injection of B16-Akaluc cells. Data are representative of two independent experiments with three mice per group and are represented as means ± SD. BLI, bioluminescence intensity; FRET, Förster resonance energy transfer; IMD, intensity-modulated display; NK, natural killer; PBS, phosphate-buffered saline.

The online version of this article includes the following figure supplement(s) for figure 5:

**Figure supplement 1.** DNAM-1-mediated ERK activation in the killer NK cells in vitro.

**Figure supplement 2.** In vivo dynamics of ERK activity in NK cells after target cell contact.

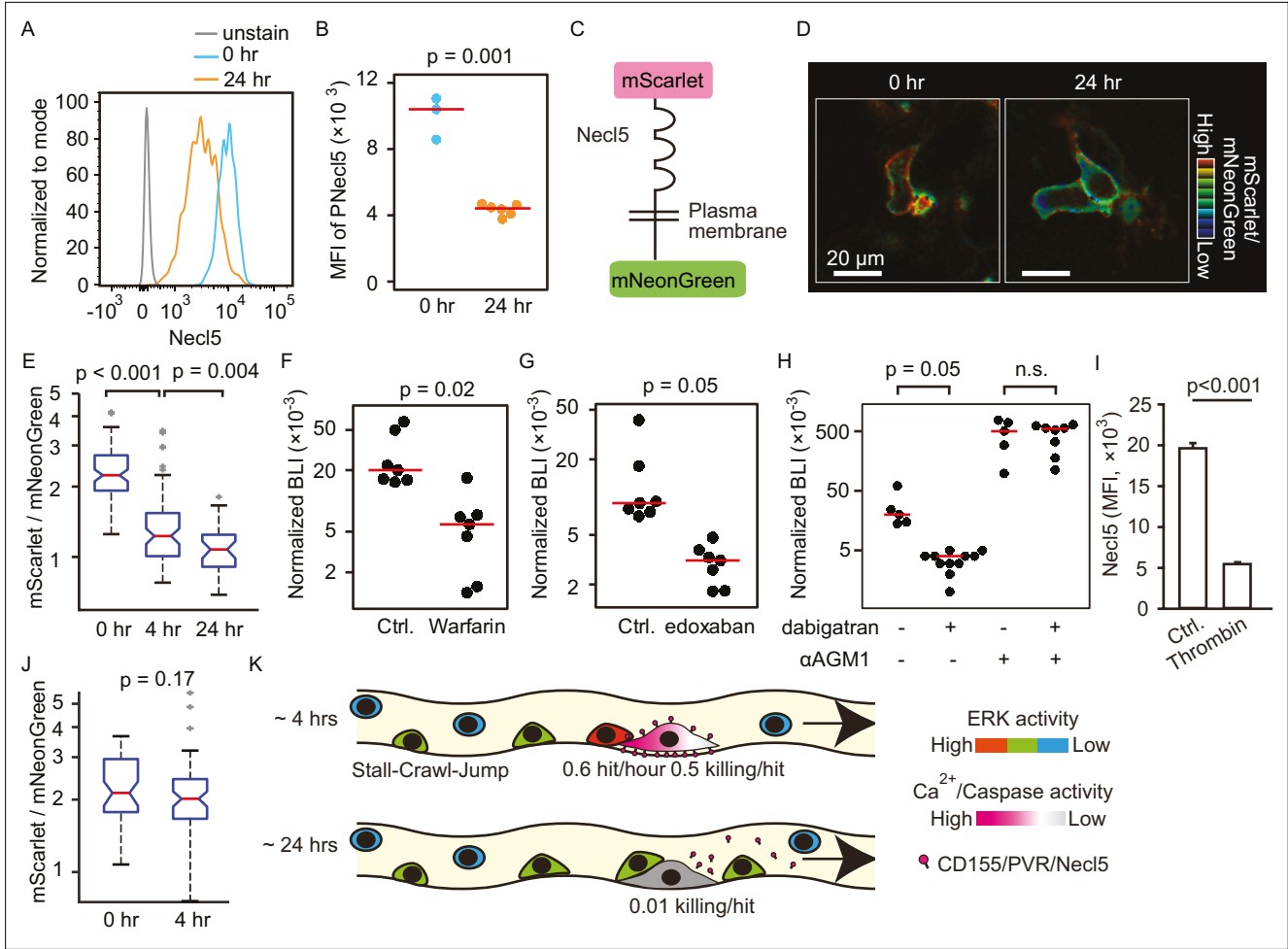

**Figure 6.** Shedding of Necl5 correlates evasion of NK cell surveillance. (**A, B**) B16-Akaluc cells were injected into the tail vein and the expression level of Necl5 on survived tumor cells was analyzed at 24 hr after dissemination. The MFI of Necl5 in tumor cells injected 0 hr or 24 hr before is shown in (**B**). Red lines represent the median. Data were pooled from two independent experiments. (**C**) Schematic representation of the Necl5-ScNeo fusion protein. (**D**) The representative images of mScarlet/mNeonGreen ratio in the B16F10 cells expressing Necl5-ScNeo at 0.5 hr and 24 hr after injection are shown in the IMD mode. The quantified mScarlet/mNeonGreen ratio in the transmembrane in indicated time point is shown in (**E**). Data were pooled from three animals. Mice were treated in their drinking water with 5 mg/L warfarin at least for 5 days and intravenously injected with 5×10⁵ B16-Akaluc cells. The BLI at 24 hr, which normalized to those at 1 hr after tumor injection for each mouse are shown. Red lines represent the median. Data were pooled from two independent experiments. (**G, H**) Mice were intravenously injected with 5×10⁵ B16-Akaluc cells. At 1 hr before and 12 hr after tumor injection, edoxaban, dabigatran etexilate, or vehicle was orally administered to mice. For NK cell depletion, mice were pretreated with either control antibody or αAGM1. The BLI at 24 hr, which normalized to those at 1 hr after tumor injection for each mouse is shown. Red lines represent the median. Data were pooled from two independent experiments. (**I**) Flow cytometric analysis of B16F10 cells treated with recombinant thrombin for 3 hr. Data are representative of two independent experiments with three wells per group and are represented as means ± SD. (**J**) Mice were orally administrated with dabigatran etexilate 1 hr before tumor injection and analyzed as in (**D, E**). Data were pooled from three animals. (**K**) Evasion of NK cell surveillance by shedding of Nec-l5. NK, natural killer.

The online version of this article includes the following video and figure supplement(s) for figure 6:

**Figure supplement 1.** Edoxaban promotes the elimination of disseminated tumor cells.

**Figure supplement 2.** Lack of micro-thrombus around the disseminated tumor cells.

**Figure supplement 3.** Elimination of *Necl5⁻/⁻ Nectin2⁻/⁻* cells.

**Figure 6—video 1.** Thrombus-formation by laser ablation around the disseminated tumor cells.

https://elifesciences.org/articles/76269/figures#fig6video1

Because the expression of Nectin2 was significantly less than that of Necl5 in B16F10 melanoma cells, we focused on Necl5. As anticipated, cell surface expression of Necl5 was markedly decreased in tumor cells isolated from the lungs 24 hr after injection (*Figure 6A and B*). To explore the basis for this change, we expressed a recombinant Necl5 fused extracellularly to the mScarlet fluorescent protein and intracellularly to the mNeonGreen fluorescent protein (*Figure 6C*). The ratio of extra-cellular mScarlet versus intracellular mNeonGreen was markedly reduced in 24 hr, indicating loss of extracellular domain, possibly by cleavage or shedding (*Figure 6D and E*). A clue to the mechanism of this Necl5 loss came from evidence that inhibition of serine proteases in the coagulation cascade has anti-tumor effects in rodent models and human patients (*Francisco and Palumbo, 2019*; *Nierodzik and Karpatkin, 2006*). Warfarin, an anti-vitamin K drug, potently accelerated the acute elimination of tumor cells (*Figure 6F*). Similarly, specific inhibitors of factor Xa and thrombin, edoxaban, and dabigatran etexilate, respectively, also promoted the elimination of tumor cells (*Figure 6G and H*). Notably, dabigatran etexilate did not have significant effect in αAGM1-treated mice (*Figure 6H*), indicating that the anti-metastatic effect is mediated by NK cells. In agreement with these functional results, thrombin cleaved off the extracellular domain of Necl5 from tumor cells in vitro (*Figure 6I*), and dabigatran etexilate suppressed the cleavage of mScarlet-tagged Necl5 in vivo (*Figure 6I and J*). Since dabigatran etexilate-treated mice often suffered from bleeding, we used edoxaban to address the long-term effect. The mice treated with edoxaban eliminated tumor cells almost completely, supporting the idea that the suppression of Necl5 cleavage promotes the elimination of disseminated tumor cells by NK cells (*Figure 6—figure supplement 1*). Several mechanisms have been proposed for coagulation cascade-mediated NK cell inhibition (*Francisco and Palumbo, 2019*), including that platelet-tumor aggregates physically protect tumor cells from NK cells (*Cluxton et al., 2019*; *Nierodzik and Karpatkin, 2006*). However, we rarely observed microthrombi around the dissemi-nated tumor cells, even when a microthrombus was easily induced by laser ablation in the vicinity of the tumor cell (*Figure 6—figure supplement 2* and *Figure 6—video 1*). Considering the role of plate-lets as the platform for coagulation factors, platelets may play a pro-metastatic role by promoting thrombin activation and thereby Necl5 shedding from the metastatic tumor cells.

Finally, we attempted to examine if the loss of Necl5 and Nectin2 on B16F10 cells cause NK cell evasion in vivo, as was the case in vitro (*Figure 5—figure supplement 1G*). Surprisingly, whereas we did not observe $Ca^{2+}$ influx after the NK cell contact with the $Necl5^{-/-}$ $Nectin2^{-/-}$ B16F10 cell (*Figure 4H*), the $Necl5^{-/-}$ $Nectin2^{-/-}$ B16F10 cells expressing AkaLuc were eliminated as efficiently as the parent B16F10 cells under the bioluminescence imaging (*Figure 6—figure supplement 3*). This observation probably indicates that an alternative pathway(s) are also involved in the $Necl5^{-/-}$ $Nectin2^{-/-}$ B16F10 cell rejection by NK cells in vivo.

## Discussion

Despite the established role of NK cells in the prevention of metastasis (*López-Soto et al., 2017*), the step(s) of the metastatic cascade at which NK cells eliminate disseminated tumor cells remain unknown. Here, we adopted the AkaBLI system, by which even a single Akaluc-expressing tumor cell could be detected in mice (*Iwano et al., 2018*), and followed the fate of intravenously injected tumor cells from 5 min to 10 days after the injection of tumor cells (*Figure 1*). In agreement with previous studies (*Grundy et al., 2007*; *Hinuma et al., 1987*), the number of tumor cells decreased rapidly in an NK cell-dependent manner. However, we noticed that as early as 24 hr after injection, tumor cells started to increase and formed macrometastatic nodules, irrespective of the presence or absence of NK cells. Our results demonstrate that the principal role of NK cells in the prevention of lung metas-tasis is to destroy tumor cells shortly after their arrival and lodging in the pulmonary vasculature, but not thereafter. Using intravital imaging, *Headley et al., 2016* found that tumor microparticles are rapidly ingested by myeloid cells to evoke an immune response (*Headley et al., 2016*). However, depletion of myeloid lineage cells did not affect the development of metastasis in the models we employed (*Figure 1—figure supplement 3*), excluding a role of the myeloid cells in the clearance of disseminated tumor cells in the immediate-early phase. Patrolling behavior similar to that of NK cells has also been observed for intravascular monocytes in the skin (*Auffray et al., 2007*). Like NK cells, the crawling monocytes require LFA-1 for their crawling. However, again, depletion of monocytes did not prevent the rapid clearance of disseminated tumor cells (*Figure 1—figure supplement 3*). Thus,

the rapid elimination of the disseminated tumor cells within 24 hr is dependent primarily on NK cells in our models.

The crawling NK cells induce pre-apoptotic calcium influx in approximately 50% of the tumor cells that they contact (*Figures 4H and 6K*), indicating that pulmonary NK cells can eliminate the disseminated tumor cells as efficiently as do chimeric antigen receptor (CAR) T cells (*Cazaux et al., 2019*). Our findings are superficially inconsistent with reports that lung NK cells exhibit highly differentiated and hypofunctional phenotypes in vitro (*Hayakawa and Smyth, 2006*; *Marquardt et al., 2017*; *Robinson et al., 1984*). This discrepancy probably arises from the fact that most of the tumor cells were killed by NK cells crawling on the pulmonary endothelial cells, but not by those flowing in the blood. NK cells adhere to the endothelial cells in an LFA-1-dependent manner (*Figure 4J*). Binding of LFA-1 to its ligand ICAM-1 induces reorganization of the actin cytoskeleton and polarization of NK cells, which is a prerequisite for subsequent redistribution of cytotoxic granules toward the bound targets (*Mace et al., 2009*). It is likely that LFA-1 engagement of ICAM on pulmonary endothelial cells contributes to the pre-activation status of NK cells. In this regard, the slow migration speed of NK cells in the presence of tumor cells may indicate the increased fraction of active NK cells (*Figure 3*).

In the B16F10 melanoma metastasis model, DNAM-1 expression in NK cells contributes to the rejection of tumor cells (*Gilfillan et al., 2008*). In agreement with a previous report (*Zhang et al., 2015*), we have shown that, upon target cell engagement, ERK is rapidly activated in DNAM-1$^+$ NK cells, but not DNAM-1$^-$ NK cells (*Figure 5—figure supplement 1D*). Simultaneous in vivo visualization of ERK activity in NK cells and Ca$^{2+}$ influx in tumor cells revealed highly efficient tumor cell killing by NK cells showing evidence of effective signaling based on ERK activation (*Figure 5E and F*). Only about 60% of lung NK cells express DNAM-1 (*Tahara-Hanaoka et al., 2005*). However, most of the tumor cells were killed by the NK cells crawling on the pulmonary endothelial cells, but not by those flowing in the blood. Because DNAM-1 serves as an adhesion molecule (*Kim et al., 2017*; *Shibuya et al., 1996*), the crawling NK cells may be biased to DNAM-1$^+$ NK cells, consistent with the potent cytotoxic activity of the crawling cells.

The discrepancy of the effect of *Necl5* and *Nectin2* knockout between in vitro and in vivo reflects the complexity of the mechanism of activation and cytotoxicity of NK cells (*Figure 5—figure supplement 1G* and *Figure 6—figure supplement 3*). First, *Gilfillan et al., 2008* observed that the number of lung metastasis was significantly increased in DNAM-1-deficient mice, but markedly less than mice injected with αAGM1, indicating that DNAM-1-Necl5 interaction is not the only mechanism of B16F10 rejection by NK cells (*Gilfillan et al., 2008*). Second, *Chan et al., 2010* reported that lung metastasis of B16F10 cells was impaired in DNAM-1 deficient mice treated with cytokines; however, neither DNAM-1 deficiency nor anti-DNAM-1 did not exhibit significant effects on the lung metastasis in the absence of cytokine administration (*Chan et al., 2010*). Because we stimulated NK cells with IL2 only in vitro, our observation agrees with this report. Third, *Li et al., 2018* reported that *Necl5*-deficient B16F10 cells form lung metastasis less efficiently than WT cells because Necl5 is critical for tumor cell migration and survival (*Li et al., 2018*). Therefore, the effect of reduced sensitivity to NK cells may be masked by the other effects of *Necl5* and *Nectin2* deficiency. Moreover, there are three receptors, DNAM-1, TIGIT, and CD96, that bind to Necl5 with different affinity, 119, 3.15, and 37.6 nM, respectively (*Martinet and Smyth, 2015*). Since the two inhibitory receptors, TIGIT and CD96, exhibit higher affinity than the activating receptor DNAM-1, thrombin-mediated reduction of Necl5 may strengthen the inhibitory signal. Finally, the discrepancy could also be explained by the previous reports that TIGIT$^+$ NK cells can eliminate *Necl5*$^{-/-}$ cells in vivo by missing-self recognition due to the education via Necl5 on the host cells (*He et al., 2017*) and that the effect of education on degranulation does not last in cell culture with IL2 (*Pugh et al., 2019*). The elimination mechanism of *Necl5* and *Nectin2* knockout cells is suggested to be NK cell-dependent but Ca$^{2+}$ influx-independent manners because we did not detect Ca$^{2+}$ influx in *Necl5* and *Nectin2* knockout cells in vivo (*Figures 4H and 6K*).

Shedding of ligands for the activating receptors of NK cells has been documented for NKG2D (*Raulet et al., 2013*). The NKG2D ligands, MICA, MICB, and ULBP2 are cleaved by matrix metalloproteases (MMPs), leading to evasion of NK surveillance. It is also reported that platelets can promote the MMP-mediated shedding of the NKG2D ligands in vitro (*Maurer et al., 2018*). However, to the best of our knowledge, involvement of thrombin or factor Xa in the shedding of the ligands for activating receptors such as Necl5 has not been reported. Interestingly, Necl5 contributes to cell adhesion as does DNAM-1 (*Takai et al., 2008*). Therefore, Necl5 on tumor cells is a double-edge sword,

which facilitates adhesion to the lung capillary but also invokes NK cell attack. To evade the NK cell surveillance, after anchoring to the capillaries, tumor cells are able to take advantage of the capacity of thrombin to strip Necl5 from the cell surface within 24 hr after adhesion (*Figure 6J*). This new understanding of how NK cell surveillance of tumor cells is regulated in the micro-circulation provides a rationale for the development of new ways to inhibit the growth of clinically significant lung metastases.

## Materials and methods

### Plasmids

Plasmids encoding R-GECO1, SCAT3, GCaMP6s, mScarlet were obtained from Takeharu Nagai (*Zhao et al., 2011*), Masayuki Miura (*Takemoto et al., 2003*), and Addgene (plasmid # 40753 for GCaMP6s; #85044 for mScarlet; Cambridge, MA), respectively. The cDNAs of mNeonGreen (*Shaner et al., 2013*) was synthesized with codon optimization by GeneArt (Thermo Fisher Scientific, Waltham, MA). A plasmid encoding tdTomato was obtained from Takara Bio (#632533; Kusatsu, Japan). pPBbsr2-Venus-Akaluc was described previously (*Iwano et al., 2018*). For the gene knockout, lentiCRISPR v2 vector (Addgene plasmid #52961) was used. Plasmids pCSIIhyg-R-GECO1 and pCSIIbsr-GCaMP6s, lentiviral vectors for R-GECO1 and GCaMP6s, respectively, were constructed by inserting cDNAs into pCSII-based lentiviral vectors (*Miyoshi et al., 1998*) with IRES-hyg (hygromycin B-resistance gene) or IRES-bsr (blasticidin S-resistance gene). psPAX2 (Addgene plasmid #12260) and pCMV-VSV-G-RSV-Rev (provided by Hiroyuki Miyoshi at RIKEN) were used for the lentivirus production. To generate pPBbsr-SCAT3-NES, cDNA coding the SCAT3 fused with the nuclear export signal (NES) (LQLPPLERLTLD) of the HIV-1 rev protein (*Fischer et al., 1995*) was subcloned into pPBbsr, a PiggyBac transposon vector with IRES-bsr (*Yusa et al., 2009*). To generate pPBbsr2-Necl5-ScNeo, cDNA coding the signal peptide of Necl5 (a.a. 1–28), mScarlet, Necl5 (a.a. 29–408), and mNeonGreen were PCR-amplified and assembled into pPBbsr2 vector by using In-Fusion system (Takara Bio USA, Inc, Mountain View, CA). pCMV-mPBase (obtained from the Wellcome Trust Sanger Institute) was co-transfected with pPB vector to establish stable cell lines. To generate pT2Aneo-tdTomato-CAAX, cDNA encoding tdTomato fused with the CAAX domain of the KRas protein (a.a. 170–189) was subcloned into pT2Aneo vector (obtained from *Kawakami et al., 2004*), a Tol2 transposon vector with IRES-neo (neomycin-resistance gene). To generate transgenic mice, pT2ADW-lox-mCherry-hyBRET-ERK-NLS was constructed by assembling cDNAs of hyBRET-ERK-NLS (*Komatsu et al., 2018*), mCherry, and the NES sequence of the HIV-1 rev protein into pT2ADW vector (*Komatsu et al., 2018*) by In-Fusion cloning (Takara Bio).

### Reagents

PD0325901 (FUJIFILM Wako Pure Chemical Corporation, Osaka, Japan) was applied as a MEK inhibitor. Warfarin, Lixianar (edoxaban), and Prazaxa (dabigatran etexilate) were obtained from Eisai Co., Ltd (Tokyo, Japan), Daiichi Sankyo Company, Limited (Tokyo, Japan), and Boehringer Ingelheim GmbH (Ingelheim, Germany), and used as a vitamin K inhibitor, factor Xa (FXa) inhibitor, and thrombin inhibitor, respectively. AkaLumine-HCl (TokeOni) was obtained from Kurogane Kasei Co., Ltd (Nagoya, Japan) or synthesized as previously described (*Kuchimaru et al., 2016*) and used as the substrate of Akaluc. Collagenase type IV and DNase I were obtained from Worthington Biochemicals (Lakewood, NJ) and Roche (Basel, Switzerland), respectively. A LIVE/DEAD Fixable Red Dead Cell Stain Kit (Thermo Fisher Scientific) or 7-AAD (BD Bioscience) was used to stain dead cells in flow cytometry. DyLight 488-labeled Lycopersicon esculentum lectin was purchased from Vector Laboratories (Burlingame, CA). Recombinant mouse protein C and factor Xa were obtained from R&D Systems, Inc (Minneapolis, MN). Recombinant mouse thrombin was obtained from antibodies-online GmbH (Aachen, Germany).

### Antibodies

The following antibodies were used for staining: BV510 or FITC anti-CD45 (30-F11), APC-Cy7 anti-CD3 (145–2C11), PerCP-Cy5.5 anti-NK1.1 (PK136), APC or PE-Cy7 anti-DNAM-1 (10E5), PE anti-F4/80 (BM8), APC anti-CD11b (M1/70), PE anti-NCR1 (29A1.4), PE anti-Ly-6G (1A8), PE anti-c-Kit (2B8), APC anti-CD49b (DX5), PE-Cy7 anti-CD200R3 (Ba13), PE anti-ICAM-1 (YN1/1.7.4), Alexa 488 anti-ICAM-2 (3C4), PE-anti PVR/Necl5 (TX56) (all from BioLegend, San Diego, CA), recombinant mouse DNAM-1

Fc chimera protein (R&D Systems), and Alexa Fluor 647 goat anti-mouse IgG (H + L) cross-adsorbed secondary antibody (Thermo Fisher Scientific). The following antibodies were used for in vivo blocking experiments: anti-LFA-1α (M17/4), anti-Mac-1 (M1/70) (both from Bio X Cell, West Lebanon, NH), and Rat IgG2a isotype control antibody (RTK2758; BioLegend). The following antibodies or reagents were used for in vivo cell depletion: anti-asialo GM1 (FujiFilm Wako Pure Chemical Corporation, Osaka, Japan), Rabbit IgG isotype control (Thermo Fisher Scientific), anti-Ly6G (1A8; BioLegend), Rat IgG2a isotype control, anti-CD200R3 (Ba103; Hycult Biotech, Uden, the Netherlands), Rat IgG2b isotype control (RTK4530; BioLegend), clodronate liposomes (Hygieia Bioscience, Osaka, Japan), or control liposome (Hygieia Bioscience).

## Tumor cells

The melanoma cell line B16F10 was purchased from the Cell Resource Center for Biomedical Research (Sendai, Japan). The MC-38 mouse colon adenocarcinoma cell line was provided by Takeshi Setoyama and Tsutomu Chiba at Kyoto University. The Braf$^{V600E}$ melanoma cell line (*Dhomen et al., 2009*) was provided by Caetano Reis e Sousa at the Francis Crick Institute. 4T1 mammary tumor cells were purchased from ATCC (Manassas, VA) and maintained on a collagen-coated dish (AGC Techno Glass, Tokyo, Japan).

## Cell culture

All cell lines were cultured in complete RPMI medium (Thermo Fisher Scientific) containing 10% fetal bovine serum (FBS) (Sigma-Aldrich, St. Louis, MO), 1 mM sodium pyruvate (Thermo Fisher Scientific), 50 µM 2-mercaptoethanol (Nacalai Tesque), 1% GlutaMAX solution (Thermo Fisher Scientific), 1% MEM non-essential amino acids (Thermo Fisher Scientific), 10 mM HEPES solution (Thermo Fisher Scientific), 50 µM 2-mercaptoethanol (Nacalai Tesque), 100 units/ml penicillin, and 100 µg/ml strep-tomycin (Nacalai Tesque, Kyoto, Japan). Mycoplasma contamination is regularly checked using Plas-moTest mycoplasma detection kit (InvivoGen, San Diego, CA).

## Establishment of stable cell lines

To prepare the lentivirus, pCSIIhyg-R-GECO1 or pCSIIbsr-GCaMP6s was cotransfected with psPAX2 and pCMV-VSV-G-RSV-Rev into Lenti-X 293T cells (Clontech, Mountain View, CA) with Polyethylen-imine 'Max' (Mw 40,000; Polysciences, Warrington, PA). Virus-containing media were harvested 48 hr after transfection, filtered, and used to infect B16F10 cells to yield B16-R-GECO cells and B16-GCaMP cells. For the transposon-mediated gene transfer, pPBbsr-SCAT3-NES, pPBbsr2-Venus-Akaluc, or pPBbsr2-Necl5-ScNeo was cotransfected with pCMV-mPBase into B16F10 cells by using Lipofectamin 3000 reagent (Thermo Fisher Scientific), yielding B16-SCAT3 cells, B16-Akaluc cells, and B16-Necl5-ScNeo cells, respectively. pT2Aneo-tdTomato-CAAX was cotransfected with pCS-TP into B16-GCaMP6 cells by using Lipofectamin 3,000 reagent, yielding B16-GCaMP-tdTomato-CAAX cells. pPBbsr2-Venus-Akaluc was cotransfected with pCMV-mPBase into Braf$^{V600E}$ melanoma cells, MC-38 cells, and 4T1 cells by using Lipofectamin 3000 reagent. Cells were selected with either 10 µg/ml blasticidin S (Calbiochem, San Diego, CA) or 100 µg/ml hygromycin B (FujiFilm Wako Pure Chemical Corporation).

## CRISPR/Cas9-mediated establishment of KO cell lines

For CRISPR/Cas9-mediated KO of *tyrosinase* (*Tyr*), *Necl5*, and *Nectin2*, single-guide RNAs (sgRNA) targeting the first or second exon were designed using the CRISPRdirect program (http://crispr.dbcls.jp/). For the establishment of double knockout cells, a puromycin-resistant gene in lentiCRISPR v2 vector was replaced with a bleomycin-resistant gene. The targeting sequences were as follows: *Tyr*, GGGTGGATGACCGTGAGTCC; *Necl5*, GCTGGTGCCCTACAATTCGAC; *Nectin2*, GACTGCGG CCCGGGCCATGGG. Annealed oligo DNAs for the sgRNAs were cloned into the lentiCRISPR v2 vector. The sgRNA/Cas9 cassettes were introduced into cells by lentiviral gene transfer. Infected cells were selected by 3.0 µg/ml puromycin (InvivoGen) or 100 µg/ml zeocin (Thermo Fisher Scientific). Cells deficient for *Necl5* and *Nectin2* were sorted by a FACS Aria IIu cell sorter and used without single-cell cloning. *Tyr*-KO cells were subjected to single-cell cloning and examined for the frame-shift mutation by nucleotide sequencing.

## Mice

C57BL/6N (hereinafter called B6) mice, BALB/c mice, and nude mice were purchased from Shimizu Laboratory Supplies (Kyoto, Japan) and bred at the Institute of Laboratory Animals, Graduate School of Medicine, Kyoto University under specific-pathogen-free conditions. B6N-Tyrc-Brd/BrdCrCrl (hereinafter called B6 Albino) mice were obtained from Charles River Laboratories. Mice of either sex were used at the age of 6–18 weeks. B6.Cg-Gt(ROSA)26Sortm9(CAG-tdTomato)Hze/J mice (JAX 007909) were obtained from the Jackson Laboratory (Bar Harbor, ME). B6(Cg)-Ncr1tm1.1(icre)Viv/ Orl mice (*Narni-Mancinelli et al., 2011*) (hereinafter called *Ncr1*iCre mice) were obtained from INFRA-FRONTIER (Oberschleissheim, Germany). Transgenic mice expressing hyBRET-ERK-NLS have been described previously (*Komatsu et al., 2018*). B6.Cg-Gt(ROSA)26Sortm9(CAG-tdTomato)Hze/J mice were crossed with *Ncr1*iCre mice for NK cell-specific expression of tdTomato, resulting in *Ncr1*iCre/ B6.Cg-Gt(ROSA)26Sortm9(CAG-tdTomato)Hze/J mice (hereinafter called NK-tdTomato mice). Tg(lox-mCherry-hyBRET-ERK-NLS) mice were crossed with *Ncr1*iCre mice for NK cell-specific expression of hyBRET-ERK-NLS, resulting in *Ncr1*iCre/ Tg(lox-mCherry-hyBRET-ERK-NLS) mice (hereinafter called NK-ERK mice). Mice of either sex were used for experiments without specific randomization and blinding. We performed at least two independent experiments with at least three mice each for the condition of interest. For the imaging experiments, if we failed to see any fluorescence or bioluminescence signal, the mice were excluded from the analysis. The animal protocols were reviewed and approved by the Animal Care and Use Committee of Kyoto University Graduate School of Medicine (Approval no. 19090).

## Generation of transgenic mice

Transgenic mice were generated by Tol2-mediated gene transfer as previously described (*Sumiyama et al., 2010*). Briefly, fertilized eggs derived from B6 mice were microinjected with a mixture of Tol2 transposase mRNA and pT2ADW-lox-mCherry-hyBRET-ERK plasmid. The offspring mice, named Tg(lox-mCherry-hyBRET-ERK-NLS), were then backcrossed with B6 Albino mice for at least three generations. Newborn mice were illuminated with a green LED and inspected for red fluorescence through a red filter LED530-3WRF (Optocode, Tokyo).

## In vivo cell depletion

To deplete NK cells, mice were injected i.p. with 20 µg αAGM1 or rabbit IgG isotype control antibody on 1 and/or 2 days before tumor cell injection. In some experiments, the antibody administration was repeated on days 0 and 7 after tumor cell injection. To deplete basophils, 30 µg αCD200R3 (Ba103) or rat IgG2b isotype control antibody was injected i.p. at 1 day before tumor cell injection. To deplete neutrophils, 200 µg anti-Ly6G (1A8) or rat IgG2a isotype control was injected i.p. at 1 day before tumor cell injection. To deplete monocytes and macrophages, clodronate liposome (50 mg/kg) or control liposome was injected i.p. at 1 day before tumor cell injection. The efficiency of the depletion was assessed by flow cytometry.

## In vivo blocking of integrins

NK-tdTomato mice were injected i.v. with 100 µg of αLFA-1α, αMac-1, or Rat IgG2a antibody during image acquisition. To visualize the vascular structure, 50 µg of DyLight 488-labeled lectin was intravenously injected. The number of NK cells in the 0.25-mm$^2$ FOV was counted 0–10 min before and 30 min after antibody injection. Image acquisition and analysis were carried out with MetaMorph software (Molecular Devices LLC, Sunnyvale, CA).

## Staining of intravascular NK cells of the bone marrow and lung

Mice were subjected to intravenous injection of the 3 µg αCD45 antibody 3 min before sacrifice. Bone marrow cells were harvested by flushing the femoral bone with 5 ml RPMI containing 10% FBS, 100 units/ml penicillin, and 100 µg/ml streptomycin. Red blood cells were removed by lysis with ACK lysing buffer and centrifugation at 500×*g* for 5 min at 4°C. A single-cell suspension of the lung cells was generated by mincing the resected lungs with scissors and incubating the minced tissue in RPMI containing 200 U/ml collagenase type IV and 5 U/ml DNase I for 30 min at 37°C. The lysed tissue was then passed through a φ40-µm cell strainer. The flow-through fraction was washed with

phosphate-buffered saline (PBS) by centrifugation at 500×*g* for 5 min at 4°C. Cells were analyzed by flow cytometry.

## Flow cytometry analysis

After staining, cells suspended in PBS containing 3% FBS were analyzed and/or sorted with a FACS Aria IIu cell sorter (Becton Dickinson, Franklin Lakes, NJ). The following combinations of lasers and emission filters were used for the detection of fluorescence: for the fluorescence of BV510, a 405-nm laser and a DF530/30 filter (Omega Optical); for the fluorescence of FITC and Alexa 488, a 488-nm laser and a DF530/30 filter; for the fluorescence of PerCP/Cy5.5 and 7-AAD, a 488-nm laser and a DF695/40 filter (Omega Optical); for the fluorescence of PE, a 561-nm laser and a DF582/15 filter (Omega Optical); for the LIVE/DEAD Fixable Red Dead Cell Stain Kit, a 561-nm laser and a DF610/20 filter (Omega Optical); for the fluorescence of PE-Cy7, a 561-nm laser and a DF780/60 filter (Omega Optical); for the fluorescence of APC, a 633-nm laser and a DF660/20 filter (Omega Optical); and for the fluorescence of APC-Cy7, a 633-nm laser and a DF780/60 filter. Cells were first gated for size and granularity to exclude cell debris and aggregates, and dead cells were excluded by LIVE/DEAD Fixable Red Dead Cell Stain Kit or 7-AAD. Data analysis was performed using FlowJo software (Tree Star, Ashland, OR). For intracellular staining of Ly6G, isolated splenocytes were fixed and permeabilized with BD Cytofix/Cytoperm Fixation/Permeabilization Solution Kit (Thermo Fisher Scientific). Then, cells were incubated with anti-Ly6G antibody for 20 min at 4°C. Cells were washed twice with BD Perm/Wash Buffer and stained with FITC Goat anti-rat IgG antibody for 20 min at 4°C. Cells were washed three times with BD Perm/Wash Buffer and further stained with anti-CD11b antibody for 20 min at 4°C. After twice wash with BD Perm/Wash Buffer, cells were analyzed by flow cytometry as described above.

## Instrumentation settings of bioluminescence imaging

Bioluminescent images were acquired using a MIIS system (Molecular Devices, Japan, Tokyo) equipped with an iXon Ultra EMCCD camera (Oxford Instruments, Belfast, UK) and a lens (MDJ-G25F095, $\phi$ 16 mm, F: 0.95; Tokyo Parts Center, Saitama, Japan). Images were acquired under the following condition: binning, 4; EM gain, 1000. For blocking of LFA-1, mice were injected with 100 µg of αLFA-1α antibody or an isotype control antibody, rat IgG2a, 2 hr before tumor injection. For MEK inhibition, mice were injected with PD0325901 i.p., 1 hr before and 8 hr after tumor injection. For vitamin K-dependent protease inhibition, mice were treated in their drinking water with 2.5–5 mg/L warfarin dissolved in an ethanol solution. Control mice were treated with the ethanol vehicle solution only. Solutions were prepared fresh every 3–4 days. The treatment started at least day –5 before tumor injection and continued until the end point of the experiment. For factor X or thrombin inhibition, mice were orally administrated with 20 mg/kg edoxaban or 330 mg/kg dabigatran etexilate twice a day. Prothrombin time was measured on blood samples collected from warfarin, edoxaban, and dabigatran etexilate-treated mice before starting experiment using CoaguChek XS system (Roche).

## Bioluminescence imaging of lung metastasis

B6 albino, BALB/c, or nude mice (female, 8–10 weeks old) were used in this experiment. Akaluc-expressing cells were suspended in PBS at $2 \times 10^6$ cells/ml, and 250 µl of the suspension was injected intravenously 60 min before imaging. Mice were anesthetized with 1.5% isoflurane (FujiFilm Wako Pure Chemical Corporation) inhalation and placed on a custom-made heating plate in the supine position. Immediately after the administration (i.p.) of 100 µl of 5 mM AkaLumine-HCl, bioluminescent images were acquired using a MIIS system (Molecular Devices, Japan, Tokyo). During the interval of long-term observation (more than 1 hr), mice were recovered from anesthesia and, immediately before observation, anesthetized again and administered AkaLumine as described. In some experiments, $4 \times 10^5$ B16F10 melanoma cells in 50 µl PBS containing 50% Geltrex (Thermo Fisher Scientific) were injected into the footpad of 6 weeks old female mice. After 2 weeks, B16-Akaluc cells were injected from tail vein. For serial injection, $5 \times 10^5$ B16-Akaluc cells were injected into the tail vein of mice that had been injected PBS or B16F10 cells into tail vein 24 hr before. For continuous observation of less than 3 hr, 100 µl of 15 mM AkaLumine-HCl was administered to the mice immediately after tumor injection. Acquisition of bioluminescent images was started at 5 min after tumor injection and repeated every

1 min. Image acquisition and analysis were carried out with MetaMorph software. The details of instrumentation settings and drug administration are summarized in Supplementary Methods.

## Bioluminescence imaging of spontaneous lung metastasis

BALB/c or nude mice were inoculated with $5\times10^5$ or $1\times10^4$ 4T1-Akaluc cells suspended in 50 µl PBS containing 50% Geltrex at the footpad of right hind limb, respectively. Every 2–3 days, bioluminescence images were acquired immediately after the administration (i.p.) of 100 µl of 5 mM AkaLumine-HCl. To mask bioluminescence signals from the primary tumor site, right hind limb was covered with black silicon clay. Images were acquired under the following condition: binning, 4 no EM gain, exposure: 180 s. Image acquisition and analysis were carried out with MetaMorph software.

## Instrumentation settings of two-photon excitation microscope and characterization of flowing and crawling NK cells

Mice were observed with an FV1200MPE-BX61WI upright 2P excitation microscope (Olympus, Tokyo, Japan) equipped with an XLPLN 25XW-MP 25×/1.05 water-immersion objective lens (Olympus) and an InSight DeepSee laser. Image areas of 500×500 µm² to a depth of 25 µm were acquired every 30 s for 60–120 min, with Z steps at 2.5 µm. Images of 512×512 pixels were scanned at 2 µs/pixel with 1.0–1.2× digital zoom. The excitation wavelengths for cyan fluorescent protein, green fluorescent protein, and tdTomato were 840, 930, and 1040 nm, respectively. We used an IR-cut filter, BA685RIF-3 (Olympus), two dichroic mirrors, DM505 and DM570 (Olympus), and four emission filters, BA460-500 (Olympus) for cyan fluorescent protein, BA495-540 (Olympus) for green fluorescent protein, BA520-560 (Olympus) for yellow fluorescent protein, and 645/60 (Chroma Technology Corp, Bellows Falls, VT) for tdTomato fluorescence. For the characterization of flowing and crawling NK cells, 1.5–2×10⁶ B16-SCAT3 cells or B16-GCaMP cells were administered to NK-tdTomato mice through the tail veil. After 2 hr, images of a 0.25-mm² FOV at a depth of 25 µm were acquired every 30 s for 2 hr. Crawling NK cells were defined as cells whose trajectory was recorded in more than four frames, that is 2 min, before contact with a melanoma cell. Flowing NK cells were defined as cells that were already in contact with the melanoma cells when they first appeared in the FOV. Cells were counted manually by using MetaMorph software. For imaging the signaling molecules activity in vivo, fluorescent images were acquired with three channels using the following filters and mirrors: an infrared (IR)-cut filter, BA685RIF-3, two dichroic mirrors, DM505 and DM570, and four emission filters, FF01-425/30 (Semrock, Rochester, NY) for the second-harmonic generation channel (SHG Ch), BA460-500 for the CFP Ch, BA520-560 for the FRET and GCaMP6s Ch. The excitation wavelength for CFP and GCaMP was 840 nm.

## Intravital pulmonary imaging by two-photon excitation microscopy

Lung intravital imaging was performed as described previously (*Kamioka et al., 2017*) with some modifications. In brief, mice were anesthetized by 1.5% isoflurane inhalation and placed in the right lateral position on an electric heating pad. The body temperature was maintained at 36°C using a heating pad with a BWT-100A rectal thermometer feedback controller (Bio Research Center, Nagoya, Japan). The mice were then anesthetized with 1.0% isoflurane supplied through a tracheostomy tube Surflo indwelling catheter 22G (Terumo, Tokyo, Japan) connected to an MK-V100 artificial respirator (Muromachi Kikai, Tokyo, Japan). The respirator condition was as follows: $O_2$ and air gas ratio, 80:20; beats per min, 55; gas flow, 35 ml/min; inspiratory/expiratory ratio, 3:2. The left lung lobe was exposed by 5th or 6th intercostal thoracotomy with custom-made retractors. A custom-made vacuum-stabilized imaging window was placed over the lung. Minimal suction (0.3–0.4 bar) was applied to stabilize the lung against the coverslip. All movies were median-filtered for noise reduction. Image analysis was carried out with Imaris (Bitplane, Belfast, UK) and MetaMorph software. The details of instrumentation settings and characterization of flowing and crawling NK cells are summarized in Supplementary Methods.

## Tracking and motion analysis of NK cells in the lung

For 3D tracking, time-lapse image areas of 500×500 µm² and 25 µm thickness at a depth of 10–35 µm were acquired every 30 s. In some experiments, 1.5–2×10⁶ B16-SCAT3 cells were intravenously injected and images were acquired for 2 hr after 4 hr after tumor injection. Image analysis was carried

out with Imaris and MetaMorph software. Tracking of NK cells or tumor cells was performed by the 3D tracking function of Imaris. The time-series data of the coordinates were used to calculate track duration, length, speed, and displacement. The parameters of 3D tracking by Imaris were as follows: max distance, 30 µm; max frame gap, 2. We used the 3D position of traced cells for the motion analysis. The instantaneous speed $v$ was calculated as the speed between two consecutive time frames, that is, $v = |r(t) - r(t - \Delta t)|/\Delta t$, where r is the position of cells, t is the elapsed time, and $\Delta t$ is the time interval. We chose $\Delta t$=0.5 min. For *Figure 3D–F*, we ensembled all data over the cells and the elapsed time up to 50 min. The MSD of NK cells was calculated by the following equation: $\text{MSD} = \left\langle |r_i(t) - r_i(0)|^2 \right\rangle_i$, where $r_i$ is the position of cell i, and ⟨⟩ represents the average over cells. For the curve fitting in the MSD analysis, we used the nonlinear least-squares solver 'lsqcurvefit,' a built-in function of MATLAB (Mathworks Inc, Natick, MA) to determine the exponent parameter of the diffusivity. In general, the MSD adopts the asymptotic power-law form: $MSD\ t^{\alpha}$, where $\alpha$ is the degree of diffusive motion. The motion is classified as normal diffusion when α=1 but as anomalous diffusion otherwise (*Krummel et al., 2016*). An NK cell hit on a tumor is defined by the event when an NK cell comes within 10 µm of a tumor cell. The hit probability is obtained by dividing the total number of hit events by the sum of the observation period of each tumor cell. MATLAB scripts and the data sets are available from the source code file.

## Visualization of signaling molecule activity in vivo

To detect caspase activity and $Ca^{2+}$ influx in tumor cells under a 2P microscope, $1.5–2.0×10^6$ B16-SCAT3 cells, B16-GCaMP cells, or B16-GCaMP-tdTomato-CAAX cells with *Tyr* deficiency were administered to NK-tdTomato mice through the tail vein. After tumor injection, a 500×500 µm² FOV at a depth of 25 µm was imaged every 30 s for 4–6 hr. For simultaneous observation of ERK activity in NK cells and $Ca^{2+}$ influx in melanoma cells, $1.5–2.0×10^6$ B16-GCaMP cells were administered to NK-ERK mice through the tail veil. After tumor injection, a 500×500 µm FOV at a depth of 25 µm was imaged every 30 s for 6 hr.

## In vitro culture of NK cells

NK cells were purified by negative selection from mice splenocytes with an NK cell isolation kit II (Miltenyi Biotec, Bergisch Gladbach, Germany) in accordance with the manufacturer's instructions. The post-sort purity of NK cells (NK1.1$^+$CD3$^-$) was >95%. Purified NK cells were plated in 96-well U-bottomed plates (Thermo Fisher Scientific) in complete RMPI medium supplemented with the recombinant murine 1000 U/ml IL-2 (PeproTech, Rocky Hill, NJ) and cultured for 5 days. In some experiments, DNAM-1$^+$ or DNAM-1$^-$ NK cells were purified by a FACS Aria Ⅱu on day 2, and further cultured for 3 days. The purity of each NK cell fraction was >98%.

## Time-lapse imaging of in vitro killing of tumor cells

B16-R-GECO cells with *Tyr* deficiency ($2×10^4$) were plated on a collagen-coated 96-well glass-base plate (AGC, Tokyo, Japan) and cultured for more than 6 hr to facilitate cell adhesion. Immediately after starting imaging with an epifluorescence microscope, $2×10^4$ NK cells derived from hyBRET-ERK-NLS mice were added to the wells containing adherent target cells. The cells were imaged with an IX81 inverted microscope (Olympus) equipped with a UPlanSApo 40×/0.95 objective lens (Olympus), a PRIME scientific CMOS camera (Photometrics, Tucson, AZ), a Spectra-X light engine (Lumencor, Beaverton, OR), an IX2-ZDC laser-based autofocusing system (Olympus), a MAC5000 controller for filter wheels and XY stage (Ludl Electronic Products, Hawthorne, NY), and an incubation chamber (Tokai Hit, Fujinomiya, Japan). The filters and dichroic mirrors used for time-lapse imaging were as follows: for FRET imaging, an 430/24 (Olympus) excitation filter, an XF2034 (455DRLP) (Omega Optical, Brattleboro, VT) dichroic mirror, and FF01-483/32 (Semrock) and 535/30 (Olympus) for CFP and FRET, respectively. For Red fluorescent protein imaging, 572/35 (Olympus) excitation filters, 89006 (Chroma Technology Corp) and FF408/504/581/667/762-Di01 (Semrock) dichroic mirrors, and 632/60 (Olympus) emission filters, respectively. MetaMorph software was used for background noise subtraction and image analysis. Background intensities were determined by using an empty culture dish with the same amount of media. After background subtraction, the FRET/CFP ratio images were represented in the intensity-modulated display (IMD) mode. In the IMD mode, eight colors from red to blue were used to represent the FRET/CFP ratio, with the intensity of each color indicating the mean

intensity of FRET and CFP channels. To track NK cells, the CFP images were analyzed by using the Fiji TrackMate plugin. From the x and y coordinates, the fluorescence intensity of each cell was isolated and the FRET/CFP ratio was calculated by MATLAB.

## Counting of macroscopic lung metastasis

Single-cell suspensions of B16-Akaluc cells ($5×10^5$) were injected intravenously into mice. The lungs were harvested on day 14 or 15, and tumor nodules were counted under a dissection microscope.

## Flow cytometric analysis of disseminated tumor cells in the lungs

$1.5–2.0×10^6$ B16-Akaluc cells were intravenously injected and single-cell suspension of the lung cells was generated after 24 hr as described above. The expression level of Necl5 was analyzed by FACS Aria IIu cell sorter. Data analysis was performed using FlowJo software.

## Quantification of shedding of extracellular domain of Necl5 in the lung

$1.5–2.0×10^6$ B16-Necl5-ScNeo cells were intravenously injected. Images were acquired after 4 hr or 24 hr by intravital pulmonary imaging with a 2P excitation microscope as described above. After 24 hr, moving to the other FOV without B16-Necl5-ScNeo cells and $1.0×10^6$ B16-Necl5-ScNeo cells were newly injected from tail vein. Images of newly injected B16-Necl5-ScNeo cells were acquired with the same condition with the image acquisition of 24 hr. To examine the effect of anticoagulants, mice were orally administrated with 330 mg/kg dabigatran etexilate 1 hr before tumor injection. The intensity of mScarlet and mNeonGreen in the plasma membrane was isolated by using MetaMorph software and the mScarlet/mNeonGreen ratio was calculated.

## In vitro protease digestion

$1.0×10^5$ B16F10 cells or 293T cells were resuspended in serum-free RPMI and incubated for 3 hr at 37°C with 100 µg/ml thrombin. The expression level of murine Necl5 was analyzed by FACS Aria IIu cell sorter. Data analysis was performed using FlowJo software.

## Observation of thrombus in pulmonary capillaries

$1.5–2×10^6$ B16-GCaMP-tdTomato-CAAX cells were administered to hyBRET-ERK-NES mice through the tail veil and images were acquired after 24 hr. During image acquisition, injuries on endothelial walls were generated by momentarily exposing a small area of the vessel wall to a laser of 70 mW power at 840 nm for up to 1 s. Image analysis was carried out with MetaMorph software.

## Quantification and statistical analysis

The statistical differences between the two experimental groups were assessed by Welch's t-test unless otherwise indicated. Kaplan-Meier survival analyses were performed using MATLAB, and the log-rank test was used to determine significance.

## Acknowledgements

The authors thank J Miyazaki, T Nagai (Osaka University), and M Miura (University of Tokyo) for the plasmids, T Setoyama, T Chiba (Kyoto University), and C Reis e Sousa (Francis Crick Institute) for the cell lines, M Yanagida (Kyoto University) for the mice, F Gochi (Kyoto University) for the intercostal thoracotomy, and R N Germain (NIAID), D A Russler-Germain (Washington University School of Medicine), and K Ikuta (Kyoto University) for the critical reading of this manuscript. The authors are grateful to the members of the Matsuda Laboratory for their helpful input, and to K Hirano, K Takakura, A Kawagishi, and Y Takeshita for their technical assistance. This work was supported by the Kyoto University Live Imaging Center. Financial support was provided by JSPS KAKENHI Grant nos. 18K15317 (to HI), 15H05949 (to MM), 16H06280 (to MM), and 19H00993 (to MM), AMED Grant no. 19gm5010003h0003 (to MM), Fugaku Trust for Medical Research (to MM), and JST CREST Grant no. JPMJCR1654 (to MM).

# Additional information

## Funding

| Funder | Grant reference number | Author |
|---|---|---|
| Japan Society for the Promotion of Science | 18K15317 | Hiroshi Ichise |
| Japan Society for the Promotion of Science | 15H05949 | Michiyuki Matsuda |
| Japan Society for the Promotion of Science | 19H00993 | Michiyuki Matsuda |
| Japan Agency for Medical Research and Development | 19gm5010003h0003 | Michiyuki Matsuda |
| Fugaku Trust for Medicinal Research | | Michiyuki Matsuda |
| Core Research for Evolutional Science and Technology | JPMJCR1654 | Michiyuki Matsuda |

The funders had no role in study design, data collection and interpretation, or the decision to submit the work for publication.

## Author contributions

Hiroshi Ichise, Conceptualization, Data curation, Formal analysis, Funding acquisition, Investigation, Methodology, Writing - original draft; Shoko Tsukamoto, Yoshinobu Konishi, Investigation; Tsuyoshi Hirashima, Data curation, Formal analysis, Writing - original draft; Choji Oki, Satoshi Iwano, Methodology, Resources; Shinya Tsukiji, Atsushi Miyawaki, Kenta Sumiyama, Methodology, Resources, Writing - review and editing; Kenta Terai, Project administration, Supervision, Writing - review and editing; Michiyuki Matsuda, Conceptualization, Data curation, Formal analysis, Funding acquisition, Project administration, Supervision, Validation, Writing - review and editing

## Author ORCIDs

Hiroshi Ichise ![ORCID] http://orcid.org/0000-0002-5187-810X
Tsuyoshi Hirashima ![ORCID] http://orcid.org/0000-0001-7323-9627
Yoshinobu Konishi ![ORCID] http://orcid.org/0000-0003-1212-7212
Shinya Tsukiji ![ORCID] http://orcid.org/0000-0002-1402-5773
Kenta Sumiyama ![ORCID] http://orcid.org/0000-0001-8785-5439
Kenta Terai ![ORCID] http://orcid.org/0000-0001-7638-3720
Michiyuki Matsuda ![ORCID] http://orcid.org/0000-0002-5876-9969

## Ethics

The animal protocols were reviewed and approved by the Animal Care and Use Committee of Kyoto University Graduate School of Medicine (approval no. 19090).

## Decision letter and Author response

Decision letter https://doi.org/10.7554/eLife.76269.sa1
Author response https://doi.org/10.7554/eLife.76269.sa2

# Additional files

## Supplementary files

• Transparent reporting form

• Source code 1. Matlab scripts and data sets.

## Data availability

Imaging data are deposited at SSBD: database (https://doi.org/10.24631/ssbd.repos.2021.08.001).

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
