## [Editor Report]

This is an interesting study that significantly contributes to our understanding of the immunological mechanisms that limit cancer metastasis. Specifically, the authors make use of advanced imaging techniques to characterize the earliest responses of NK cells against tumor cells in cancer metastasis.

---

## [Decision Letter]

[Editors' note: this paper was reviewed by Review Commons.]

---

## [Author Response]

Reviewer #1 (Evidence, reproducibility and clarity (Required)):The authors aimed to understand the control and the elimination of disseminated tumor cells by NK cells within the lung, their main question being how pulmonary NK cells are able to prevent tumor cells from colonization in the lung.To dissect this question, Hiroshi Ichise and colleagues took advantage of the ultra-sensitive bioluminescence whole body imaging system combined with intravital two-photon microscopy technology involving genetically-encoded biosensors tumor or NK cells to explore the behavior and functional competences of NK cells in an experimental lung metastasis model.First, the authors have monitored the fate of intravenously injected B16-Akaluc cells from 5 min to 10 days and observe that tumor cells decrease rapidly within the first 12-24 hours. In parallel, they performed asialoGM1+ and NK1.1+ cells depletion by injection of depleting anti-aGM1 and antiNK1.1 antibodies in order to see the involvement of these populations on the elimination of the disseminated tumor cells. They conclude that a rapid decrease of the tumor cells is mediated by NK cells. Consisting with this first data, the authors observe also the same early NK cells mediated impact on two other syngeneic mouse tumor cell lines : the BRAFV600E melanoma and the colon adenocarcinoma MC-38.In a second part, the authors dissected NK cell dynamic behaviors in the pulmonary capillaries by taking advantage of the NKp46iCRExrosa26dtTomato mice where NKp46+ cells are fluorescents and performed 2P intravital imaging to follow the in situ the NKp46+ cells behavior. They could nicely observe that NK cells arrive from the capillaries and patrol on the lung epithelial cells in a stall-crawl-jump manner. Moreover, they also show that the attachment to the pulmonary capillaries is mediated by LFA-1. In the presence of B16F10 tumor model, they observe that NK cells stay longer in the capillaries and increase their duration time of crawling indicating that NK cells stay in contact longer with tumor cells.The authors then explored the NK-mediated tumor killing in the lung by measuring tumor cell apoptosis using B16F10-SCAT3 cells (which leads to visualize caspase 3 activation) and Ca^2+^ influx in tumor cells expressing two Ca^2+^ sensors, GCaMP6s and R-GECO. They could observe casp3 activation but also Ca+ influx on tumor cells within few minutes after encountering NK cells. They also observe that evasion of NK cell surveillance is mediated by Nectin-5 and Nectin-2 expressed on tumor cells.Then, they focus on NK cell activation by looking at ERK activation. To do so, they have isolated NK cells from Tg mice expressing a FRET-based ERK biosensor and performed in vitro killing assay against B16-R-GECO tumor cells but also in vivo experiments. For the in vivo experiments, they have developed reporter mice whose NK cells express the FRET biosensor for ERK. They observe that ERK-dependent NK cell activation contributes to the elimination of disseminated tumor cells within the first few hours but not after 24 hours. Indeed, theu observe that B16F10-Akaluc tumor cells are equally eliminated when injected 24 hrs after a first injection of B16F10 or PBS in mice. The authors concluded that tumor cell acquire the capacity to evade NK cell surveillance after 24 hrs rather than a hypothesis toward NK cells loose tumoricidal activity over time.Finally, the authors have explored their last result on the potential tumor cell evasion of the NK cell surveillance. They show that this NK cell evasion is mediated by the shedding of cell surface NeCl^-^5. They next show that clivage of extracellular domain of NeCl^-^5 was mediated by thrombin in vitro and that anti-coagulation factors such as Warfarin, Edoxaban or Dabigatran Etexilate promote tumor elimination as observed by the bioluminescence experiments. This loss prevents the NK cell signaling needed for effective killing of tumor targets.However, most of the results remain correlations and have not been formally demonstrated or miss controls.B16F10 is a well-known and characterized NK cell target in a in vivo model so the first part is not really knew except the in situ behavior of NK cells within the lung capillaries. The new mechanism of thrombin-mediated shedding of NeCl^-^5 causing evasion from NK Cell surveillance is really concentrated on the last figure (Figure N{degree sign}6) and some supplemental experiments are mandatory and needed to really confirm this affirmation.

We deeply appreciate the reviewer’s effort to evaluate our work. The reviewer criticizes that the mechanism is well known except “the in situ behavior of NK cells within the lung capillaries.” Indeed, this is what we wish to show in our work. Nobody has ever seen how NK cells kill metastatic tumor cells in the lung. There is a big gap between in vitro tissue culture experiments and in vivo macroscopic counting of metastatic nodules. Most researchers do not even know when and where in the lung NK cells kill metastatic tumor cells. Live imaging is a powerful approach to address such questions.

Reviewer #1 (Significance (Required)):There are several points to address to improve the significance of these data.Major points:1. A global point : 3 mice/group is to small to analyse and interprete data because of the heterogeneity of the mice. Mean +/- SEM have to represented instead of SD.

For the sake of animal welfare, researchers are asked to use minimal number of mice. Moreover, only one mouse can be observed in each imaging session, which takes several hours. In most experiments we performed two independent experiments with three mice each. We believe the number is appropriate for this type of experiment. In case of small number of samples, we think SD is better than SEM.

2. The authors used the well-known polyclonal anti-asialoGM1 Ab to deplete NK cells. AsialoGM1 is also expressed by ILC1, T, NKT and gd+T cells but also basophils (Trambley J et al., Asialo GM1(+) CD8(+) T cells play a critical role in costimulation blockade-resistant allograft rejection. JCI, 1999). The authors checked the involvement only for the basophils. They have to check the depletion of each of these populations specifically in the lung to assume that the depletion impact only the NK cells or they must change their conclusion on the entire manuscript and say that not only NK cells is responsible and involved in the control of the disseminated tumor cells but maybe also ILC1, NKT and or gd+T cells.

We obtained similar observations by using BALB/c *nu/nu* mice, which lack T cells (Figure 1D-1G). Therefore, we can exclude the contribution of NKT and γδT cells, which are absent in BALB/c *nu/nu* mice, in the acute phase (< 24 hrs) examined in this study. ILC1 is known to promote lung metastasis of B16F10 cells {Gao, 2017 #284}, neglecting its role in the protection of metastasis in the acute phase. These statements have been included in the result section with the reference.

(Lines 167) “The experiment with *nu/nu* BALB/c mice also excluded the involvement of NKT and γδ T cells, with which αAGM1 cross-reacts (Trambley et al., 1999). […] But, ILC1 is known to promote lung metastasis of B16F10 cells (Gao et al., 2017), suggesting that the effect of αAGM1 is primarily caused by the depletion of NK cells.”

3. Lines 133 to 136 : The authors say that they “did not observe any significant difference in the relative increase of the bioluminescence signal between the control and αAGM1-treated mice, implying that NK cells eliminate disseminated melanoma cells primarily in the acute phase (< 24 hrs) of lung metastasis”. Please comment because the depletion of asGM1+ cells impact also the growth of the tumor until 8 days (Figure 1B-E-G).

After 24 hrs, the slope of increment of bioluminescence intensity (BLI) did not change significantly betweenαAGM1-treated mice and control mice. In both mice, the doubling times of melanoma cells are approximately one day. We have included this statement for the clarity (line 139-142). Importantly, after 24 hrs, we did not observe any significant difference in the relative increase of the bioluminescence signal between the control and αAGM1-treated mice. In both mice, the doubling time of melanoma cells are approximately 1 day, implying that NK cells eliminate disseminated melanoma cells primarily in the acute phase (< 24 hrs) of lung metastasis.

(Lines 137) “In both mice, the doubling time of melanoma cells are approximately 1 day, implying that NK cells eliminate disseminated melanoma cells primarily in the acute phase (< 24 hrs) of lung metastasis.”

*4. Figure S3A-B : The authors say that basophils express aGM1 so they performed basophils involvement on the elimination of B16F10 tumor cells with depleting aCD200R3 mab. They also checked the involvement of neutrophils and monocytes. They observed that basophils, neutrophils and monocytes are not involved on the B16F10 elimination. But what is the hypothesis to assess the role of neutrophils and monocytes ? Moreover, they did not explore Basophil roles in the other models including MC-38, BRAFV600E and 4T1 tumor cells*.

We depleted neutrophils and monocytes because antibody-mediated removal of leukocytes could have non-specifically increased the survival of tumor cells. We wished to show the specificity of our approach. As for expanding the number of experiments with different cell lines, we are afraid but it is too much burden, considering the period required for the experiments and animal welfare.

*(5a) Figure 1D : Missing control : the author must add the WT Balbc + a-AGM1 as contro*l.

We performed the suggested experiment. Because this experiment was conducted as a distinct set of experiment, we moved the WT BALB/c data from Figure 1D to Figure S2C. Legend to figures have been modified accordingly.

We have rephrased the sentences as follows:

(Lines 147) “We extended this approach to other syngeneic mouse tumor cell lines: Braf^V600E^ mutated melanoma cells (Dhomen et al., 2009), MC-38 colon adenocarcinoma cells (Rosenberg et al., 1986), and BALB/c mice-derived 4T1 breast cancer cells (hereinafter called 4T1-Akaluk).”

(5b) Lines 154 to 156 : the authors say that “T cell immunity does not contribute to tumor cell reduction” because tumor cells are eliminated in the nu/nu mice as efficiently as in the WT Balbc mice. This is not correct because they are looking in a window that correspond to innate immunity activation (up to 24 hrs) so they cannot talk about T cell immunity, the adaptive response will come more later around 8 days after.

Yes, we are focusing on the early phase of the rejection of metastatic tumor cells. We have rephrased the sentences as follows:

(Line 162) “T cell immunity does not contribute to tumor cell reduction in the acute phase of rejection (< 24 hrs).”

6) Line 159 : (refer to point #2) To affirm that NK cells is critical and involved in the elimination of the disseminated tumor, authors have to perform experiment in a model of NK cell deficiency. The most relevant nowadays is the NKp46ICRExrosa26DTA mice that are deficient in NK cells but also ILC1 cells. Indeed, the authors have used the NKp46iCre mice model for other questions.

As the reviewer stated, the contribution of NK cells in the rejection of metastatic tumors is very well known. Therefore, we would like to refrain from repeating the experiments by using other genetically modified mouse lines, which will take at least one year. We wish to emphasize again that the new findings of our paper are related to in vivo imaging.

(7a) Figure 2F : IC missing.

We have included the IgG2a control antibody in Figure 2F. Figure legend and the Materials and methods section have been modified accordingly.

*(7b) Lines 181-182 : Authors conclude that the effect of anti-LFA-1 on NK cells adhesion to the pulmonary endothelial cells is mediated primarily by LFA-1. It is not totally true because it is partially mediated as observed in the Figure 2F. So authors should change their conclusion and precise that the involvement is partially mediated by LFA-1*.

According to the reviewer’s suggestion, we have toned down the conclusion as follows:

(Lines 190) As expected, αLFA-1α, but not αMac-1, markedly reduced the number of NK cells on the pulmonary endothelial cells (Figure 2F), indicating that the adhesion of NK cells to the pulmonary endothelial cells is mediated at least partially by LFA-1.

8) Figure S5B-C-D and S7: The authors talk about tumor cell death. But they are analyzing Ca^2+^ influx in vitro so it is a little bit different from the cell death. I'm wondering how the cell death is measured especially in the Figure S5D and S7?

Under microscopes, apoptosis can be easily recognized by the appearance of blebs. We have modified the sentence and included a video (Video EV4) in the revised paper.

(Lines 232) “A surge of Ca^2+^ influx was observed only in cells that were doomed to die (Figure S5C and Video EV4). 98% of cells that exhibited Ca^2+^ influx died by apoptosis with blebbing (Figure S5D).”

9) Figure 4H and lines 232-233 : the authors conclude that “damage to tumor cells is dependent on the engagement of DNAM-1 on NK cells”. There is any experiment performed to affirm this point so the authors cannot maintain this conclusion. First, the authors only analyzed Ca^2+^ influx at a specific time point. So this result only show that Nectin-5 and/or Nectin-2 expressed by B16F10 is involved in the Ca^2+^ influx following NK cell contact but there is any data on DNAM-1 contribution. So, the role on the NK cells and specifically DNAM-1+ NK cells have not been addressed here. To answer to that question, the author have to perform in vivo model of engrafted WT vs NeCl^-^5/2 ko B16F10 in a WT vs DNAM1 deficient NK cells mouse model to ascertain the contribution of NeCl^-^5/2-DNAM-1 on NK cells. Moreover, survival curve and bioluminescent experiments would be very appreciated.

Re: The use of Ca^2+^ as a surrogate marker for cell death.

We showed that most, if not all, cells died after Ca^2+^ influx in vitro (Figure S5D). Due to the limitation of the observation period of in vivo imaging, we used Ca^2+^ influx as the surrogate marker of cell death. We stated the reason as follows:

(Lines 234) “With these in vitro data in hand, we used the surge of Ca^2+^ influx as the surrogate marker for apoptosis of metastatic melanoma cells in vivo.”

Re: Necl-5/Nectin-2 knockout B16F10 cells.

According to the suggestion, in these three months we worked hard to see whether we could find a condition in which, *Necl-5^-/-^ Nectin-2^-/-^* B16F10 cells evade NK cell killing in the lung. We established several *Necl-5^-/-^ Nectin-2^-/-^* B16F10 cells clones to see the effect. As anticipated, these *Necl-5^-/-^ Nectin2^-/-^* B16F10 cell clones were less sensitive to IL2-stimulated NK cells in vitro (Figure S7G). However, in the bioluminescence assay we found that *Necl-5^-/-^/Nectin-2^-/-^* B16F10 cells disappeared as rapidly as the parent B16F10 cells (Figure S10). We confirmed that there is no other DNAM-1 ligand in Necl5^-/-^/Nectin-2^-/-^ B16F10 cells by using recombinant DNAM-1 (Figure S7E) and that ERK activation in NK cells was not observed by these mutants (Figure S7F). In conclusion, our data is consistent with previous reports that examined the role of DNAM-1- Necl-5 interaction by using B16F10 cells. We have revised the result section to include the data and added a paragraph on the discrepancy between in vitro and in vivo observation.

(Lines 340)

“Finally, we examined if loss of Necl-5 and Nectin-2 on B16F10 cells cause NK cell evasion. The *Necl5^-/-^ Nectin-2^-/-^* B16F10 cells survived more efficiently than wild type B16F10 cells from charge of the IL2-stimulated NK cells in vitro (Figure S7G). […] This observation agrees with previous note that DNAM-1- Necl-5 interaction is not the only mechanism of B16F10 rejection by NK cells in vivo (Gilfillan et al., 2008).”

(Lines 393)

“The discrepancy of the effect of *Necl-5* and *Nectin-2* knockout between in vitro and in vivo reflects the complexity of the mechanism of activation and cytotoxicity of NK cells (Figure S7G and Figure S10). […] Finally, the discrepancy could also be explained by the previous reports that TIGIT^+^ NK cells can eliminate *Necl-5^-/-^* cells in vivo by missing-self recognition due to the education via Necl-5 on the host cells (He et al., 2017) and the effect of education on degranulation does not last in cell culture with IL-2 (Pugh et al., 2019).”

Re: Use of DNAM1 deficient NK cells.

We have shown in vitro data with DNAM-1-negative NK cells in Figure S7D. This experiment was performed by sorting of DNAM-1-negative NK cells. I understand the importance of the experiment with the DNAM-1-deficient mice. But the introduction of another knockout mouse line cannot be performed easily. Instead, we have toned down the conclusion on the requirement of signaling from Necl-5/Nectin-2 to DNAM-1.

(Lines 244) “As anticipated, tumor Ca^2+^ influx was almost completely abolished in the *Necl-5* and *Nectin-2-*deficient B16F10 cells (Figure 4H), suggesting that damage to tumor cells is dependent on the engagement of Necl-5 and/or Nectin-2 on melanoma cells.”

Revision of summary and abstract:

Because the data of Necl-5^-/-^/Nectin-2^-/-^ B16F10 cells did not corroborate our proposal that DNAM1-dependent killing of B16F10 cells in vivo, we rephrased our title as follows. We intend to claim that the novelty of our work is in the functional visualization of NK cell killing of tumor cells. “Functional Visualization of NK Cell-mediated Killing of Metastatic Single Tumor Cells”.

10) Lines 253-254 : the authors talk about tumor apoptosis but they are looking at Ca^2+^ influx. So, they should change their conclusion or show killing experiment.

In Figure S7, we have shown that the sustained Ca^2+^ influx is a useful surrogate marker for apoptosis. We have included this information explicitly in the revised paper.

(Lines 232) “A surge of Ca^2+^ influx was observed only in cells that were doomed to die (Figure S5C); 98% of cells that exhibited Ca^2+^ influx died by apoptosis with blebbing (Figure S5D). With these in vitro data in hand, we used the surge of Ca^2+^ influx as the surrogate marker for apoptosis of metastatic melanoma cells in vivo.”

11) Figure 6 : the authors conclude that the trombin dependent shedding of Necl-5 causes evasion of NK cells surveillance. Moreover, all experiments are correlations and do not implicate in the same experiment Necl-5, DNAM-1+ NK cells and trombin or anti-coagulation factors. So, as in the comment #9, to adress this point, the authors should inject WT vs Necl-5 deficient B16F10-Akaluc into WT vs NK cell depleted mice and monitor the bioluminescence of the tumor cells within 24 hrs following injection of anti-coagulation factors as in the Figure 6H. Moreover, the monitoring of the survival curve and the number of the lung metastasis would be also very important and informative to really answer to this point.

Because we failed to see the difference between wild-type and *Necl-5^-/-^ Nectin-2^-/-^* B16F10 cells in the in vivo bioluminescence assay, we did not perform this experiment. As already stated, IL2-stimulated NK cells recognize Necl-5 and Nectin-2-deficient B16F10 cells poorly with cytotoxicity reduced by over 40%. This 40% change may not be detected in vivo setup. Therefore, we have toned down our conclusion and suggested the presence of alternative pathway to eliminate tumor cells in the lung as already described.

Minor points:1) Figure 2E: The authors assess the involvement of LFA-1 and MAC-1 on the NK cells attachment to the pulmonary endothelial cells. But there is other adhesion molecules that are known to be expressed by NK cells as for example CR4 (CD11c/CD18). So, the attachment of NK cells could be also due to this molecule.

We agree. This question is related to the Major point #6. We have toned down the sentence as already described.

2) Lines 190 to 197 : Authors should put this methodology part in the “material and method” in order to be more clear on the message they want to deliver.

According to the suggestion, we moved the description on normal and anomalous diffusive motion to the Method Section (Supplementary material line 382).

3) Line 228 : There is any hypothesis or explanation regarding the use of Necl5/Necl2 deficient B16F10. Why authors decided to go and explore this pathway ? Authors could add some transition sentence and explanation to help readers.

According to the suggestion, we have included following statement.

(Lines 240) “It is reported that DNAM-1 on NK cells contributes to the elimination of B16F10 melanoma cells”.

4) The author could performed the same experiment as in Figure S7D and assessed ERK activation of DNAM+ vs – NK cells against WT vs Necl-5/ Necl-2KO R-GEKO B16F10 cells.

We have performed the requested experiment as a part of reply to comment #9 (Figure S7G).

5) Line 283 : Thanks to reformulate the sentence. Check the figures associated with the text.

We have read through the paper and amended errors. We apology the mistakes.

Reviewer #2 (Evidence, reproducibility and clarity (Required)):The authors use in vivo imaging techniques to investigate the killing of lung metastasis by NK cells. They demonstrate that the cleavage of CD155 may result in resistance of killing by NK cells and suggest that this could be an immune evasion mechanism of metastatic tumor cells.Overall, the subject is highly relevant, and the in vivo imaging is an interesting and highly relevant technique. However, the message, that tumor cells escape the killing by NK cells by cleavage of CD155 is interesting, but not yet fully supported by the data.

We would like to thank the reviewer for the effort of evaluation our work and very useful comments. We spent these three months to consolidate the evidence to show that the cleavage of Necl-5/CD155 contributes to the resistance of tumor cells against NK cells, but we failed to obtain positive results, probably because of the presence of an alternative pathway(s). Therefore, we have toned down our conclusion and modified the title as follows. Because the data of Necl-5-/-/Nectin-2-/- B16F10 cells did not corroborate our proposal that DNAM1-dependent killing of B16F10 cells in vivo, we rephrased our title as follows. We intend to claim that the novelty of our work is in the functional visualization of NK cell killing of tumor cells.

“Functional Visualization of NK Cell-mediated Killing of Metastatic Single Tumor Cells”.

Major comments:1. Figure 6: To support their main claim the authors would need to transfect the tumor cells with a CD155 mutant, which cannot be cleaved by Thrombin and show that these tumor cells can no longer escape NK cell-mediated killing. This experiment is straight forward and feasible. Another important experiment along this line would be the use the CD155/CD112 deficient tumor cells (Which the authors use in figure 4) in the experiments shown in figure 1. One would expect that tumor control by NK cells within the first 24 hrs is absent when using these tumor cells.

The potential thrombin cleavage site in Necl-5/CD155 is PRGSR (aa 129-133), wherein R/G is the cleavage site. We made 5 mutants, R130A, R130E, R130Q, G131W, and G131D. The G131W and G131D mutants were not detected by FACS, probably due to misfolding. The rest of 3 mutants were expressed as high as the WT Necl-5/CD155, DNAM-1^+^ NK cells failed to kill those mutants in vitro. Thus, it appears the recognition site by thrombin may also be critical for the binding to DNAM-1. Therefore, we were not able to conduct the requested experiment.

Re: Necl-5/Nectin-2 knockout B16F10 cells. Reviewer #1 raised a similar question; therefore, we repeat the same statement.

According to the suggestion, in these three months we worked hard to see whether we could find a condition in which, *Necl-5^-/-^ Nectin-2^-/-^* B16F10 cells evade NK cell killing in the lung. We established several *Necl-5^-/-^ Nectin-2^-/-^* B16F10 cells clones to see the effect. As anticipated, these *Necl-5^-/-^ Nectin2^-/-^* B16F10 cell clones were less sensitive to IL2-stimulated NK cells in vitro (Figure S7G). However, in the bioluminescence assay we found that *Necl-5^-/-^/Nectin-2^-/-^* B16F10 cells disappeared as rapidly as the parent B16F10 cells (Figure S10). We confirmed that there is no other DNAM-1 ligand in Necl5^-/-^/Nectin-2^-/-^ B16F10 cells by using recombinant DNAM-1 (Figure S7E) and that ERK activation in NK cells was not observed by these mutants (Figure S7F). In conclusion, our data is consistent with previous reports that examined the role of DNAM-1- Necl-5 interaction by using B16F10 cells. We have revised the result section to include the data and added a paragraph on the discrepancy between in vitro and in vivo observation.

(Lines 340)

“Finally, we examined if loss of Necl-5 and Nectin-2 on B16F10 cells cause NK cell evasion. The *Necl5^-/-^ Nectin-2^-/-^* B16F10 cells survived more efficiently than wild type B16F10 cells from charge of the IL2-stimulated NK cells in vitro (Figure S7G). […] This observation agrees with previous note that DNAM-1- Necl-5 interaction is not the only mechanism of B16F10 rejection by NK cells in vivo (Gilfillan et al., 2008).”

(Lines 393)

“The discrepancy of the effect of *Necl-5* and *Nectin-2* knockout between in vitro and in vivo reflects the complexity of the mechanism of activation and cytotoxicity of NK cells (Figure S7G and Figure S10). […] Finally, the discrepancy could also be explained by the previous reports that TIGIT^+^ NK cells can eliminate *Necl-5^-/-^* cells in vivo by missing-self recognition due to the education via Necl-5 on the host cells (He et al., 2017) and the effect of education on degranulation does not last in cell culture with IL-2 (Pugh et al., 2019).

2. Figure 5: The demonstration that ERK is activated in this in vivo setting is novel. However, ERK activation is not DNAM-1 specific and the ERK inhibitor is significantly less effective that the depletion of NK cells. Therefore, the relevance of these data to the main message of the manuscript is unclear and the figure could be omitted.

We agree that the modest effect of MEKi implies that activation of the DNAM-1- ERK pathway is not the sole mechanism of tumoricidal activity of NK cells. Indeed, the result described above show that there is an alternative pathway(s) in vivo. However, we wish to emphasize that ERK activation is a useful marker of NK cell activation. The data shown here vividly show the timing of NK cell activation and following tumor cell killing. Because the in vivo dynamics of NK cell activation and tumor cell killing is the most important message of this work, we wish to show this data. We have modified the conclusion as follows:

(Lines 285) “Although interpretation is limited by the action of the soluble inhibitor on cells other than NK cells, these additional data are consistent with our imaging data and the idea that activated NK cell contributes to the elimination of disseminated tumor cells and that ERK activation can be used a marker for activated NK cells.”

3. In general, the issue of NK cell exhaustion should be addressed in more detail. The experiments do not address serial killing activity of NK cells and more data is needed to show that it is not an exhaustion of NK cells but the cleavage of CD155 from the tumor cells that prevents further killing.

We believe that NK cell exhaustion would not happen in a population level in this metastasis model by the following reason. There are 2.4 million NK cells in the lung and 1.0 million NK cells are replaced every minute (Supplementary Table). Therefore, NK cells outnumber 0.5 million melanoma cells injected into circulation. Moreover, each tumor cell in the lung is hit by an NK cell every two hours. Even if we assume that melanoma cells can inactivate NK cells every time, exhaustion of lung NK cells would not happen in 12 hours. Therefore, if NK cell exhaustion could happen, it should be mediated by indirect mechanism such as cytokine-induced suppression or endothelial cell-mediated inactivation etc. We have included these statements in the result section.

(Lines 304) “These results imply that NK cells in the lung are exhausted in 24 hrs. Our quantitative data shows that 2.4 million NK cells in the lung outnumber 0.5 million melanoma cells injected into circulation (Supplementary Table). Therefore, if NK cell exhaustion could happen, it is not caused by the killing of tumor cells, but by an indirect mechanism such as cytokine-induced suppression or endothelial cell-mediated inactivation.”

*Figure 1C: The relevance of this experiment needs to be better explained.*

We have modified the result section as follows:

(Lines 139) “In tumor burden mice, the lung microenvironment is often reprogrammed in favor of metastasis (Altorki, Markowitz et al., 2019). To test this possibility, B16F10 melanoma cells were inoculated into the foot pad two weeks prior to the intravenous injection of the B16-Akaluc cells (Figure 1C).”

2. Figure 3A: What does SHG stand for?

We have included the statement that SHG stands for second harmonic generation channel in the figure legend (Line 794).

3. Figure 3: Please add a statistical analysis for these experiments.

We have included P values in the Figure 3 of the revised paper.

4. Figure 4: The use of the caspase-3 and the calcium sensors may detect different cytotoxic mechanisms used by the NK cells. While caspase-3 can be activated by death receptor and perforin/granzyme B mediated killing, the calcium sensor may report mostly on perforin mediated membrane damage. These killing mechanisms have different kinetics and are differentially used during serial killing by NK cells. This should be addressed (at least in the discussion).

We thank this invaluable comment. We have included this point as follows:

(Lines 220) “During serial killing of tumor cells by NK cells, perforin/GrzB initially plays a major role, followed by Fas-mediated killing (Prager et al., 2019). Therefore, we used two biosensors to detect NK cell-mediated killing in vivo: SCAT-3 for Fas-mediated caspase activation and GCaMP-6s for the perforin/GrzB-mediated membrane damage.”

Reviewer #2 (Significance (Required)):Investigating the in vivo cytotoxicity of NK cells against tumor cells by using live imaging technologies is highly relevant for the understanding of the dynamic relationship between tumor and killer cells. Therefore, the subject of this manuscript and the technologies used are very relevant, as in vivo killing activities do not always translate to the in vivo setting.

We thank the reviewer for the favorable comment.

Reviewer #3 (Evidence, reproducibility and clarity (Required)):Summary:Ichise et al., present a solid work describing the modality and time frame of action of NK control over seeding metastatic cells within the lung vasculature. Th authors use a variety of technique able to dissect how NK patrol lung vasculature, that they interact with cancer cells as they interact with the endothelial cells and they activate a ERK dependent activation leading to calcium influx in cancer cells leading to their death. The data support the notion that this NK control occur over an early time frame, 4h after cancer cells arrival and is mediated by Necl expression on cancer cells. After this time point cancer cells show a thrombin dependent loss of Necl expression on their surface and therefore become resistant to NK control.Comments:The data presented are supporting the conclusions. This work utilizes a variety of elegant strategy combining reporter strategy with in vivo imaging to assess the phenomenon of interaction, ERK activation, Calcium Inflax and Apoptosis activation directly in the lung.In term of experiments, I found the work thorough and complete.The data a presented well overall and the statistics seems adequate. I only have few suggestions:

We would like to thank the reviewer for the favorable comments.

Supplementary Figure S3, show the use of antiLy6G to deplete neutrophils in the lungs of C57BL/6 mice injected with melanoma B16F10 cells. It was recently shown that this antibody is not efficient in depleting neutrophils in this background, but only lead neutrophils to internalise the Ly6G so they cannot be detected by FACS. As shown in Boivin et al. 2020 http://doi.org/10.1038/s41467-020-165969) neutrophils depletion in C57BL/6 mice can be achieved by using antiGr1 antibody. Therefore, if the authors aim to show this additional control, which I also agree is really good to have, I suggest performing the experiment accordingly to the best-known practice.

According to the suggestion, we have examined intracellular Ly-6G (Figure S3E, lower panels) and found that it was decreased from 1.54% to 0.68%. So, we have toned down the effect of αLy-6G as follows:

(Lines 154) “Similarly, the roles of circulating monocytes and neutrophils were examined with clodronate liposome and αLy-6G neutrophil antibody, respectively. […] Although the effect of these reagent to eliminated monocytes and neutrophils was not complete, these data suggest the involvement of monocytes and neutrophils in the rejection of melanoma cells in the acute phase.”

Figure 1E: in the text the experiment is described as 4T1 Akaluc cells were inoculated into the foot pad of BALB/c mice with either control antibody or αAGM1, but the legend states that mice subcutaneously injected with B16 Akaluc cells into footpad. As B16 melanoma cells are not in BALB/c background, I assume the legend needs to be corrected as the cells should be 4T1, however I wonder if injecting 4T1 breast cancer cells in the footpad could have let to the substantial growth required for lung metastasis without impairing the animal mobility. Could it be that cells where actually injected in the fat pad of the mice and this is just a misspelling in the text?

We apology the erratum in the legend. We used 4T1-Akaluc cells in the experiment shown in Figure 1E, because 4T1 cells inoculated into footpad can be spontaneously metastasized to the lung (Kamioka et al., 2017). The legend has been amended accordingly.

(Lines 764) “€ BALB/c mice were pre-treated with either control antibody or αAGM1. Shown are representative merged images of the bright field and the bioluminescence images of mice subcutaneously injected with 5×10^5^ 4T1-Akaluc cells into footpad.”

In this case, the different in the tissue residence NK cells could also potentially explain why 4T1 are not cleared in the fat pad like the B6 cells are in the footpad.

We cannot neglect this possibility. But, the αAGM1 treatment did not affect the growth of the primary tumor as described (Figure 1F), suggesting that probably the effect of local NK cells is modest.

The authors should comment on the difference in the in clearance of the cells at the injection site in Figure 1C VS Figure 1E.

In both subcutaneous injection model, tumor cells at the injection site were not eliminated. Although we show it in the 4T1 subcutaneous injection model in Figure 1F but not in the B16F10 model. We have included the following sentence:

(Lines 142) “In this model, B16F10 melanoma cells continue to grow until at day 14 when B16Akaluc cells were further injected.”

Reviewer #3 (Significance (Required)):The present work is highly relevant to the field of cancer metastasis. While it is known that NK are responsible for the first line of defence against metastatic seeding, most of the studies focuses on how they are suppressed or influenced by other immune cells. The present study provides a very accurate description of their mechanism of action, how they depend in the interaction with the endothelial cells and highlight the novel aspect of thrombin in inducing cancer cells NK resistance. What cause thrombin activation is the next relevant question, by in my opinion this study is complete and important.My field of expertise is cancer metastasis and their interaction with the immune system and I personally enjoy very much reading this work.

We thank the reviewer for favorable comments and appreciate the effort to evaluate our work.

Other modifications:

1. Affiliation has been modified.

2. ORCID IDs have been updated.

3. Acknowledgement: We have referred to Dr. D. A. Russler-Germain, who provided comments (line 531).

4. Materials and methods: nectin cell adhesion molecule 2 (*Nectin-2*) was changed to *Nectin-2*. Line 499

5. Median was changed to mean. Line 221

6. A part of the Materials and methods section was moved to Supplementary Methods.

7. Materials and methods section was modified to fit the Author Checklist.